# Endosomal dysfunction contributes to cerebellar deficits in spinocerebellar ataxia type 6

**Anna A Cook[1], Tsz Chui Sophia Leung[1], Max Rice[1,2], Maya Nachman[1], Élyse Zadigue-Dube[1], Alanna Jean Watt[1]\***

[1]Biology Department, McGill University, Montreal, Canada; [2]Department of Biological Sciences, Columbia University, New York, United States

**\*For correspondence:**
alanna.watt@mcgill.ca

**Competing interest:** The authors declare that no competing interests exist.

**Abstract** Spinocerebellar ataxia type 6 (SCA6) is a rare disease that is characterized by cerebellar dysfunction. Patients have progressive motor coordination impairment, and postmortem brain tissue reveals degeneration of cerebellar Purkinje cells and a reduced level of cerebellar brain-derived neurotrophic factor (BDNF). However, the pathophysiological changes underlying SCA6 are not fully understood. We carried out RNA-sequencing of cerebellar vermis tissue in a mouse model of SCA6, which revealed widespread dysregulation of genes associated with the endo-lysosomal system. Since disruption to endosomes or lysosomes could contribute to cellular deficits, we examined the endo-lysosomal system in SCA6. We identified alterations in multiple endosomal compartments in the Purkinje cells of SCA6 mice. Early endosomes were enlarged, while the size of the late endosome compartment was reduced. We also found evidence for impaired trafficking of cargo to the lysosomes. As the proper functioning of the endo-lysosomal system is crucial for the sorting and trafficking of signaling molecules, we wondered whether these changes could contribute to previously identified deficits in signaling by BDNF and its receptor tropomyosin kinase B (TrkB) in SCA6. Indeed, we found that the enlarged early endosomes in SCA6 mice accumulated both BDNF and TrkB. Furthermore, TrkB recycling to the cell membrane in recycling endosomes was reduced, and the late endosome transport of BDNF for degradation was impaired. Therefore, mis-trafficking due to aberrant endo-lysosomal transport and function could contribute to SCA6 pathophysiology through alterations to BDNF–TrkB signaling, as well as mishandling of other signaling molecules. Deficits in early endosomes and BDNF localization were rescued by chronic administration of a TrkB agonist, 7,8-dihydroxyflavone, that we have previously shown restores motor coordination and cerebellar TrkB expression. The endo-lysosomal system is thus both a novel locus of pathophysiology in SCA6 and a promising therapeutic target.

## eLife assessment

This manuscript provides **valuable** insights to the underlying mechanism for Spinocerebellar ataxia 6 (SCA6) due to defective endolysosomal trafficking of BDNF and its receptor TrkB. The findings are **compelling** and significant in understanding the underlying pathology of SCA6. The authors have acknowledged the experimental weaknesses and recognize there may be multiple mechanisms to explain the findings.

## Introduction

Spinocerebellar ataxia type 6 (SCA6) is an inherited ataxia which presently has no cure (*Solodkin and Gomez, 2012*). It is caused by a CAG expansion mutation that generates an expanded polyglutamine

**Figure 1.** Endo-lysosomal genes are dysregulated in the spinocerebellar ataxia type 6 (SCA6) cerebellum. (**a**) Schematic showing the endo-lysosomal system. (**b**) Heat map showing relative expression of endo-lysosomal-associated transcripts in the cerebellum of five wildtype (WT) and five SCA6 mice (1 column per mouse). Genes are separated into groups based on their gene ontology categorization (GO term Endosome and Lysosome; the group 'Endolysosome' denotes genes that belong to both GO term categories), only genes with an adjusted p value <0.05 are shown.

The online version of this article includes the following source data for figure 1:

**Source data 1.** Normalized reads of endolysosomal genes.

tract in the translated protein (*Zhuchenko et al., 1997*). The affected gene is *CACNA1A* (*Zhuchenko et al., 1997*), which encodes both the alpha subunit of voltage-gated calcium P/Q channels and a putative transcription factor (*Du et al., 2019*; *Du et al., 2013*). The cellular dysfunction giving rise to disease in SCA6 is poorly understood. Here, we examine the role of the endo-lysosomal system in SCA6 pathophysiology.

The endo-lysosomal system comprises of a series of membrane-bound compartments that sort and transport endocytosed molecules destined for recycling or degradation (*Figure 1a*; *Hu et al., 2015*; *Huotari and Helenius, 2011*). These compartments are highly dynamic and regularly interact and fuse with each other as cargo is sorted and trafficked. Each compartment has a characteristic pH mediated

by ATP-dependent acidification (*Hu et al., 2015*), and is characterized by the presence of specific proteins that allow them to carry out their functions. Rab GTPases regulate trafficking of cargo, and each compartment is associated with a specific signature of Rab family members (*Stenmark, 2009*). Endocytosed cargo arrives first at the early endosomes (*Ludwig et al., 1991*), where the slightly acidic pH leads to dissociation of ligands from their receptors. Here, the cargo will undergo sorting: those that are destined for recycling such as receptors, will be returned to the cell surface. Cargo that is destined for further sorting or degradation will move to the late endosomes, which are characterized by an increasingly acidic pH and increased enzymatic activity (*Hu et al., 2015*; *Huotari and Helenius, 2011*). The lysosome is the final destination of cargo that is to be degraded. This compartment is the most acidic and contains over 50 hydrolases that carry out rapid degradation of cargo (*de Duve, 2005*; *Lawrence and Zoncu, 2019*).

Endosome dysfunction plays a role in the pathophysiology of several neurological diseases. A notable example is Alzheimer's disease, where postmortem brain tissue from patients displays increased early endosome size and accumulation of cargo (*Cataldo et al., 2004*; *Cataldo et al., 2000*; *Cataldo et al., 1997*; *Tate and Mathews, 2006*). Similar early endosome changes are seen in Down syndrome (*Cataldo et al., 2000*) and Niemann Pick C (*Jin et al., 2004*; *Kim et al., 2023*). In the cerebellum, endo-lysosomal dysfunction can lead to cerebellar dysfunction (*Akizu et al., 2015*; *Mannan et al., 2004*; *Sterling et al., 2023*; *Vanier, 2010*), and has been identified in cerebellar diseases with diverse disease-causing mechanisms (*Alves et al., 2014*; *Zhu et al., 2022*). As endosomes are the site of disassociation of ligands from their receptors, mediating both the recycling of receptors back to the cell surface and the sorting of ligands for further signaling or degradation, dysfunction can disrupt signaling pathways due to mis-trafficking of signaling molecules.

Previously, lysosomal accumulation of protein aggregates has been observed in a mouse model of SCA6 (118Q) that features a hyperexpanded polyglutamine repeat (*Unno et al., 2012*). This suggests that lysosomes may play a role in SCA6. Cerebellar pathology in this model was worsened when lysosomal function was impaired by the loss of the lysosomal enzyme cathepsin B (*Unno et al., 2012*). However, it has been unknown whether endo-lysosomal pathway dysregulation is implicated in the natural disease progression of SCA6, or whether other endosomal compartments may be involved.

We have recently found that deficits in the brain-derived neurotrophic factor (BDNF)–tropomyosin kinase B (TrkB)–Akt signaling pathway contribute to SCA6 pathophysiology (*Cook et al., 2022*), with both BDNF and TrkB expression reduced in the cerebellar vermis. However, the underlying causes of the reduced levels of BDNF and TrkB remained unknown. Since both BDNF and TrkB are dynamically trafficked through the endo-lysosomal system (*Moya-Alvarado et al., 2023*; *Moya-Alvarado et al., 2018*; *Yamashita and Kuruvilla, 2016*), we wondered whether dysfunction in this system could underlie BDNF and/or TrkB mislocalization, contributing to subsequent cellular dysfunction in SCA6.

To address whether alterations in the endo-lysosomal system give rise to cellular dysfunction in SCA6, we examined endo-lysosomal compartments in the cerebellum of male and female SCA6$^{84Q/84Q}$ mice (referred to as SCA6 mice) in the early stages of SCA6 progression. In addition to observing abnormalities in endosomal compartments, we found that BDNF and its receptor TrkB accumulated in early endosomes instead of undergoing normal sorting and recycling. These abnormalities were present in SCA6 mice between 7 and 12 months of age, coincident with the onset of motor impairment and long before Purkinje cell degeneration (*Jayabal et al., 2015*). These changes therefore represent an early driver of SCA6 pathology. We were able to rescue the deficits in early endosomes with chronic administration of the TrkB agonist 7,8-dihydroxyflavone (7,8-DHF), a treatment that we previously found could also rescue behavioral and Purkinje cell firing deficits in the SCA6 mice. Our findings argue that endo-lysosomal dysfunction is a novel locus of pathophysiology in spinocerebellar ataxia and a potential therapeutic target to investigate further.

## Results

The endo-lysosomal system (*Figure 1a*) is critical for proper cellular function and its dysregulation is associated with mis-trafficking of important cargo. Here, we address whether endo-lysosomal dysfunction contributes to SCA6, using a mouse model (SCA6$^{84Q/84Q}$ mice, referred to as SCA6 mice) that robustly models the mid-life onset of motor deficits and cerebellar dysfunction seen in SCA6 patients (*Jayabal et al., 2015*; *Watase et al., 2008*). We have previously shown that around 7 months of age, SCA6 mice develop deficits in motor coordination that progressively worsen (*Jayabal et al., 2015*;

*Jayabal et al., 2016*) and display cellular deficits, including aberrant Purkinje cell firing and disrupted BDNF–TrkB signaling (*Cook et al., 2022*; *Chang et al., 2022*; *Jayabal et al., 2016*). The cellular alterations that lead to the reduction of BDNF and TrkB in the SCA6 cerebellum remained unknown, so we conducted further investigation into potential causes.

We carried out RNA-sequencing on cerebellar vermis tissue from 12-month-old litter-matched wildtype (WT) and SCA6 mice (*N* = 5 for each genotype) during the course of another investigation. This sequencing revealed that many genes involved in the endo-lysosomal system, as defined by the gene ontology (GO:CC) terms 'Endosome' (GO:0005768) and 'Lysosome' (GO:0005764), were significantly dysregulated in the SCA6 cerebellum (*Figure 1b*, $p_{adj} < 0.05$, see *Table 1* for a full list). The expression levels of these genes were altered by between 10% and 200% in SCA6 compared to WT. Endosomal genes were found to be both up- and downregulated in SCA6, but the genes covered by the Lysosome GO term were mostly upregulated compared to WT. The affected genes are associated with multiple endosomal compartments, meaning that the cellular consequences could be wide reaching. We therefore aimed to determine whether different compartments of the endo-lysosomal system (*Figure 1a*) are altered in the cerebellum of SCA6 mice, and whether this could contribute to previously described BDNF and TrkB deficits, via mis-trafficking through the endo-lysosomal system.

## Early endosomes are enlarged and accumulate BDNF and TrkB in Purkinje cells in SCA6 mice

Early endosomes are the first stop for newly endocytosed cargo. There, cargo undergoes sorting as the slightly acidic pH facilitates ligand release from receptors (*Figure 2a*). We carried out immunohistochemistry (IHC) against early endosome antigen 1 (EEA1) on sections of cerebellar vermis from 7- to 8-month WT and SCA6 mice and found that EEA1 staining is present in the cerebellum, particularly in Purkinje cells (*Figure 2b*). We imaged Purkinje cells and could identify large tubular early endosomes in Purkinje cell somas of both WT and SCA6 mice (*Figure 2b, c*). The early endosome marker in SCA6 Purkinje cells covered a larger area, showing that the early endosome compartment is enlarged in SCA6 (*Figure 2c,d*). A similar enlargement of early endosomes has been seen in other neurological diseases (*Cataldo et al., 2000*; *Cataldo et al., 1997*; *Jin et al., 2004*). Using the EEA1 marker to delineate early endosomes, we were able to analyze BDNF and TrkB localized within the early endosome compartment. There was a higher amount of BDNF staining within the early endosomes of Purkinje cells from SCA6 mice (*Figure 2e, f*). This was intriguing as we have previously shown that BDNF levels in the cerebellar vermis as a whole are decreased (*Cook et al., 2022*), so BDNF accumulates specifically in this compartment despite depressed levels overall in the cerebellum. We found the same was true for TrkB, as TrkB immunoreactivity was increased within the EEA1-positive early endosome compartment in SCA6 Purkinje cells compared to WT (*Figure 2g, h*). Nearly all of the TrkB staining within SCA6 Purkinje cells colocalized with the EEA1 marker (*Figure 2g*, arrowheads), whereas the tissue from WT mice showed abundant TrkB staining outside the early endosome compartment of Purkinje cells (*Figure 2g*, arrowheads). Thus, early endosomes are enlarged and accumulate BDNF and TrkB in SCA6, despite decreased levels of both proteins in the SCA6 cerebellum as a whole. This cargo mis-trafficking could lead to a decrease in the amount of free BDNF available to signal, as well as the amount of TrkB recycled to the cell surface, thereby contributing to previously described BDNF–TrkB signaling deficits in SCA6 (*Cook et al., 2022*).

## Recycling endosomes are morphologically normal in SCA6

Since we identified an accumulation of TrkB in early endosomes, we wondered whether this would affect the amount of TrkB recycled back to the cell surface. Recycling endosomes return receptors to the plasma membrane for further signaling (*Figure 3a*; *O'Sullivan and Lindsay, 2020*). Syntaxin 12/13 (Stx13) is involved in recycling of endocytosed proteins back to the plasma membrane and is located in recycling endosomes and some tubular protrusions from early endosomes that are also involved in recycling (*Prekeris et al., 1998*). We used two different antibodies for Stx13 as a marker for the recycling endosome compartment. Staining with both antibodies revealed a large network of small recycling endosomes and protrusions in Purkinje cells (*Figure 3b*, *Figure 3—figure supplement 1*). Analysis of the staining with both antibodies showed that recycling endosome area was unchanged in SCA6 Purkinje cells (*Figure 3c*, *Figure 3—figure supplement 1*). We noted colocalization between TrkB and Stx13, showing TrkB being recycled back to the cell surface after dissociation from bound

**Table 1.** List of differentially expressed genes (DEGs) in Endosome and Lysosome pathways.

| Gene name | Gene ID | Base mean | Log$_2$ fold change | Adjusted p value |
|---|---|---|---|---|
| H2-K1 | ENSMUSG00000061232 | 8985.033 | 0.557 | <0.001 |
| Cst7 | ENSMUSG00000068129 | 66.028 | 1.634 | 0.010 |
| Cd68 | ENSMUSG00000018774 | 761.094 | 0.557 | 0.006 |
| Nsg1 | ENSMUSG00000029126 | 118,453.175 | −0.214 | 0.041 |
| H2-Q7 | ENSMUSG00000060550 | 461.700 | 0.761 | 0.001 |
| Nsg2 | ENSMUSG00000020297 | 17,203.283 | 0.363 | 0.012 |
| Ctsd | ENSMUSG00000007891 | 33,856.962 | 0.382 | <0.001 |
| 2610002M06Rik | ENSMUSG00000031242 | 7455.780 | −0.157 | 0.038 |
| Prkcd | ENSMUSG00000021948 | 15,130.466 | −0.240 | 0.012 |
| Rab38 | ENSMUSG00000030559 | 74.954 | 1.387 | 0.042 |
| Ctss | ENSMUSG00000038642 | 8398.508 | 0.350 | <0.001 |
| Tmem25 | ENSMUSG00000002032 | 16,856.246 | −0.150 | 0.035 |
| Apoe | ENSMUSG00000002985 | 294,918.992 | 0.315 | 0.003 |
| Unc13d | ENSMUSG00000057948 | 1948.570 | −0.298 | 0.025 |
| Cst3 | ENSMUSG00000027447 | 72,853.485 | 0.250 | 0.046 |
| Arl8a | ENSMUSG00000026426 | 23,966.848 | −0.112 | 0.040 |
| Cd63 | ENSMUSG00000025351 | 16,922.196 | 0.306 | 0.003 |
| H2-Q4 | ENSMUSG00000035929 | 1712.362 | 0.637 | <0.001 |
| H2-Q6 | ENSMUSG00000073409 | 605.404 | 0.610 | 0.017 |
| Slc11a1 | ENSMUSG00000026177 | 351.078 | 0.472 | 0.006 |
| Tinagl1 | ENSMUSG00000028776 | 1872.544 | 0.444 | 0.013 |
| Gaa | ENSMUSG00000025579 | 42,288.649 | 0.148 | 0.046 |
| Npc2 | ENSMUSG00000021242 | 13,576.144 | 0.252 | 0.005 |
| Cpq | ENSMUSG00000039007 | 3671.835 | 0.301 | 0.043 |
| Naga | ENSMUSG00000022453 | 5578.299 | 0.171 | 0.009 |
| Man2b2 | ENSMUSG00000029119 | 5625.125 | 0.227 | 0.008 |
| Gfap | ENSMUSG00000020932 | 49,633.449 | 0.670 | <0.001 |
| Hexb | ENSMUSG00000021665 | 11,580.391 | 0.325 | 0.001 |
| Lipa | ENSMUSG00000024781 | 3011.385 | 0.339 | 0.003 |
| Meak7 | ENSMUSG00000034105 | 1681.762 | 0.255 | 0.012 |
| Ifi30 | ENSMUSG00000031838 | 755.863 | 0.547 | <0.001 |
| Trip10 | ENSMUSG00000019487 | 1776.641 | 0.312 | 0.029 |
| Borcs8 | ENSMUSG00000002345 | 2644.624 | −0.392 | <0.001 |
| Myo7a | ENSMUSG00000030761 | 5484.873 | 0.552 | <0.001 |
| Syt7 | ENSMUSG00000024743 | 66,668.649 | −0.232 | 0.001 |
| Gusb | ENSMUSG00000025534 | 2558.973 | 0.185 | 0.017 |
| Rnf19b | ENSMUSG00000028793 | 28,209.761 | −0.243 | 0.038 |
| Ctsc | ENSMUSG00000030560 | 1554.587 | 0.610 | 0.003 |
| Laptm5 | ENSMUSG00000028581 | 2659.155 | 0.278 | 0.006 |

*Table 1 continued on next page*

*Table 1 continued*

| Gene name | Gene ID | Base mean | Log$_2$ fold change | Adjusted p value |
|---|---|---|---|---|
| *Acp5* | ENSMUSG00000001348 | 57.183 | 1.434 | 0.004 |
| *Inpp5f* | ENSMUSG00000042105 | 25,096.641 | −0.158 | 0.005 |
| *Rusc1* | ENSMUSG00000041263 | 8435.506 | −0.146 | 0.044 |
| *Pld1* | ENSMUSG00000027695 | 1492.635 | 0.305 | 0.013 |
| *Ticam2* | ENSMUSG00000056130 | 58.459 | 1.110 | 0.010 |
| *Rassf9* | ENSMUSG00000044921 | 702.197 | 0.316 | 0.017 |
| *Mr1* | ENSMUSG00000026471 | 1753.049 | 0.342 | 0.006 |
| *Ece1* | ENSMUSG00000057530 | 21,877.619 | −0.213 | 0.007 |
| *Sh3gl3* | ENSMUSG00000030638 | 1097.889 | 0.255 | 0.046 |
| *Ptprf* | ENSMUSG00000033295 | 6447.473 | 0.311 | 0.032 |
| *Washc2* | ENSMUSG00000024104 | 71,355.731 | −0.263 | 0.033 |
| *Syndig1* | ENSMUSG00000074736 | 15,931.471 | −0.199 | 0.019 |
| *Pld4* | ENSMUSG00000052160 | 865.562 | 0.329 | 0.046 |
| *Ackr4* | ENSMUSG00000079355 | 668.906 | 0.813 | <0.001 |
| *Unc93b1* | ENSMUSG00000036908 | 1732.922 | 0.268 | 0.014 |
| *Rab11a* | ENSMUSG00000004771 | 12,908.215 | −0.146 | 0.028 |
| *Rnd2* | ENSMUSG00000001313 | 14,349.243 | 0.181 | 0.030 |
| *H2-Ab1* | ENSMUSG00000073421 | 394.199 | 0.802 | 0.005 |
| *Pacsin1* | ENSMUSG00000040276 | 65,047.878 | −0.166 | 0.017 |
| *Ackr3* | ENSMUSG00000044337 | 1329.920 | 0.292 | 0.032 |
| *Snap25* | ENSMUSG00000027273 | 334,660.707 | −0.225 | 0.005 |
| *Map2k1* | ENSMUSG00000004936 | 25,321.319 | −0.117 | 0.046 |
| *Cd22* | ENSMUSG00000030577 | 81.223 | 0.856 | 0.003 |
| *Nrp1* | ENSMUSG00000025810 | 1247.448 | 0.283 | 0.042 |
| *Kcnj11* | ENSMUSG00000096146 | 7056.627 | −0.188 | 0.025 |
| *Fcgrt* | ENSMUSG00000003420 | 4913.386 | 0.239 | 0.013 |
| *Dgkh* | ENSMUSG00000034731 | 13,465.118 | −0.264 | 0.022 |
| *Bsg* | ENSMUSG00000023175 | 98,426.046 | 0.221 | 0.048 |
| *Rnf128* | ENSMUSG00000031438 | 4114.551 | −0.181 | 0.016 |
| *Stambpl1* | ENSMUSG00000024776 | 2576.502 | −0.196 | 0.023 |
| *Cpne6* | ENSMUSG00000022212 | 2074.575 | 0.537 | 0.003 |
| *Ccr5* | ENSMUSG00000079227 | 417.390 | 0.454 | 0.030 |
| *Dagla* | ENSMUSG00000035735 | 19,804.120 | −0.336 | 0.012 |
| *Zfyve21* | ENSMUSG00000021286 | 2110.324 | 0.191 | 0.021 |
| *Dner* | ENSMUSG00000036766 | 118,667.368 | −0.372 | <0.001 |
| *Grb14* | ENSMUSG00000026888 | 2737.028 | 0.223 | 0.030 |
| *Scoc* | ENSMUSG00000063253 | 12,593.242 | −0.778 | <0.001 |
| *Mrc1* | ENSMUSG00000026712 | 489.484 | 0.466 | 0.045 |
| *Epn3* | ENSMUSG00000010080 | 4581.906 | 0.446 | 0.009 |

*Table 1 continued on next page*

*Table 1 continued*

| Gene name | Gene ID | Base mean | Log$_2$ fold change | Adjusted p value |
|-----------|---------|-----------|---------------------|------------------|
| *Gfra1* | ENSMUSG00000025089 | 1244.452 | 0.557 | 0.006 |
| *Tpd52l1* | ENSMUSG00000000296 | 1312.485 | 0.471 | 0.033 |
| *Vcam1* | ENSMUSG00000027962 | 3713.941 | 0.429 | <0.001 |
| *Spns2* | ENSMUSG00000040447 | 9562.590 | −0.159 | 0.029 |

BDNF in the early endosomes. The TrkB staining within the recycling endosome compartment marked by Stx13 was decreased in Purkinje cells in SCA6 mice (*Figure 3d*). Combined with the increased levels of TrkB in the early endosomes of Purkinje cells in SCA6 (*Figure 2h*), these data argue that TrkB is stuck in early endosomes in SCA6, rather than being recycled to the cell surface to the same extent as in WT Purkinje cells. This mislocalization may contribute to the depressed levels of BDNF–TrkB signaling previously described in SCA6 (*Cook et al., 2022*).

## Late endosomes

Early endosomes undergo maturation into late endosomes in order to transport cargo destined for degradation to the lysosomes (*Huotari and Helenius, 2011*; *Figure 4a*). Neurons use this degradative pathway as a way to regulate BDNF levels (*Evans et al., 2011*). We used Rab7 as a marker for late endosomes (*Vitelli et al., 1997*), and identified numerous late endosomes in Purkinje cells of both WT and SCA6 mice (*Figure 4b*). The Rab7 staining covered a smaller area in the Purkinje cells of SCA6 mice compared to WT (*Figure 4c*), indicating a reduction in the size or number of late endosomes. Most of the BDNF staining within the Purkinje cells did not colocalize with Rab7 (*Figure 4b*), which was expected as we had previously found high levels of colocalization of BDNF with the early endosome marker EEA1 (*Figure 2e*). The BDNF staining that was present within the Rab7-positive late endosome compartment was significantly reduced in SCA6 compared to WT (*Figure 4d*), suggesting that BDNF accumulates in the early endosomes earlier in the endosomal pathway (*Figure 2f*), preventing its progression through the late endosomal stage. Late endosomes transport cargo to lysosomes for degradation, so less BDNF may reach the lysosome in SCA6. This could have implications for the regulation of BDNF within Purkinje cells, as neurons tightly regulate BDNF levels via degradative pathways (*Evans et al., 2011*).

## Lysosomal morphology is unchanged in SCA6

We next aimed to determine whether lysosomes were altered in SCA6, using a dual strategy to visualize the lysosomes in cerebellar slices. We first used a pH-sensitive dye, LysoTracker Red DND 99, to visualize the acidic lysosomes (*Figure 5a, b*; *Chazotte, 2011*). As LysoTracker labels all acidic compartments, it is not sufficient to confirm the identity of lysosomes. This may be particularly problematic if other intracellular compartments undergo inappropriate acidification, which has been observed in other neurological diseases (*Pescosolido et al., 2021*). Therefore, after staining acute cerebellar slices with LysoTracker, we performed IHC for Lysosomal-associated membrane protein 1 (Lamp1), a glycoprotein localized at the membranes of lysosomes. Because even Lamp1 can be localized outside of lysosomes (*Cheng et al., 2018*), we used both LysoTracker and Lamp1 to identify lysosomes in Purkinje cells. We found that both stains labelled lysosomes within Purkinje cells (*Figure 5b*). As expected, the Lamp1 staining appeared as ring-like structures, marking the membranes of the lysosomes, while LysoTracker staining marked the acidic lumen of the lysosomes, in the center of these rings (*Figure 5b*, see zoom). While Lamp1 staining identified a large number of lysosomes of varying size, the LysoTracker stain was present only in the largest. This discrepancy has been noted previously, as LysoTracker does not label the smallest lysosomes in cerebellar slices or other cell preparations that are fixed after staining (*Chazotte, 2011*; *Song et al., 2008*). We found that the total area covered by the Lamp1 and LysoTracker stains was not significantly different between Purkinje cells from WT and SCA6 mice (*Figure 5c, d*). The area of colocalization between the two stains was also unchanged (*Figure 5e*).

To investigate further and determine whether the two stains label the same population of lysosomes in the two genotypes, we counted the number of lysosomes labeled by different combinations of the

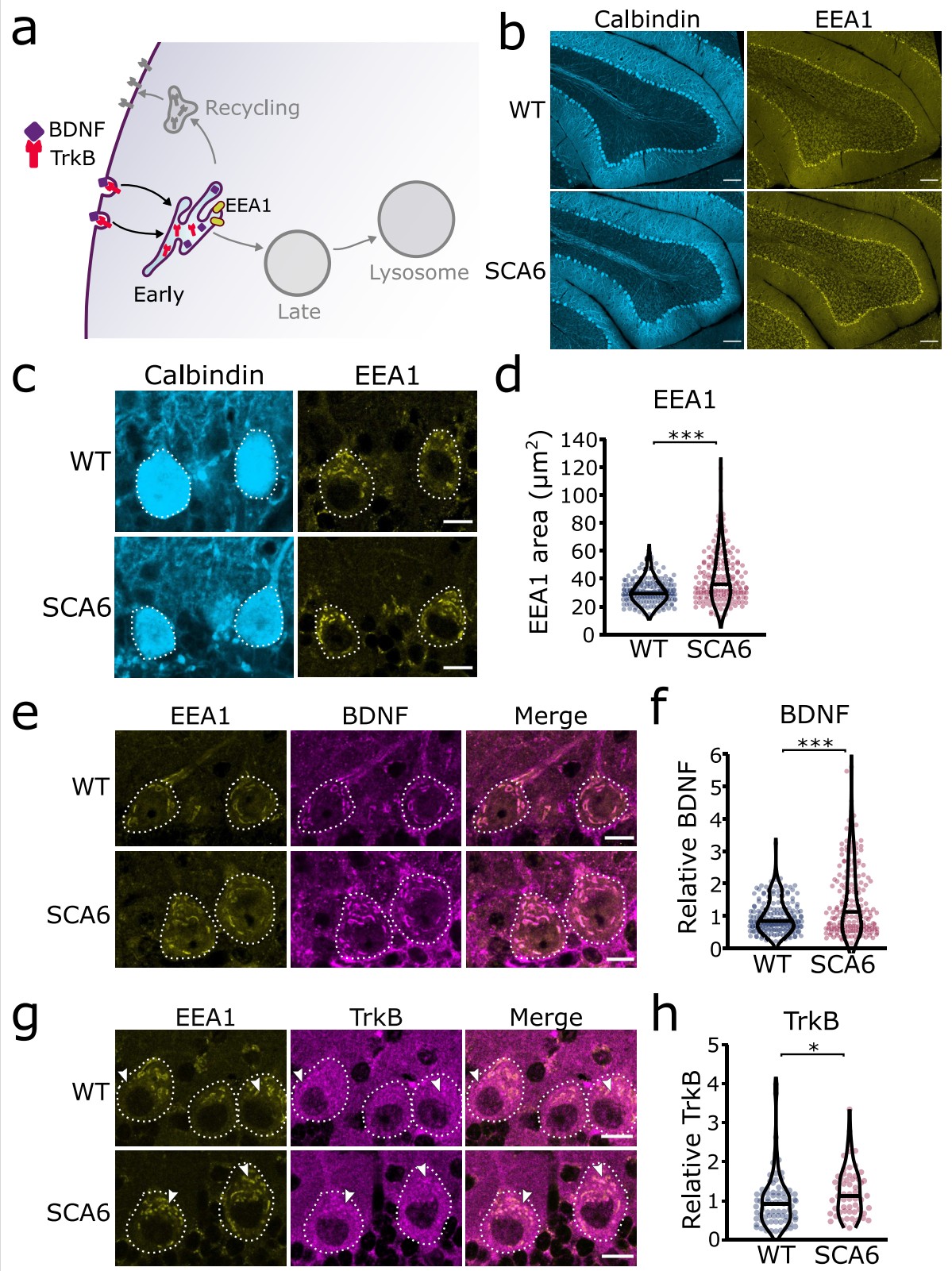

**Figure 2.** Early endosomes are enlarged and accumulate brain-derived neurotrophic factor (BDNF) and tropomyosin kinase B (TrkB) in spinocerebellar ataxia type 6 (SCA6) Purkinje cells. (**a**) Schematic showing BDNF and TrkB endocytosis and trafficking to early endosomes labeled with early endosome antigen 1 (EEA1). (**b**) The early endosome marker EEA1 stains early endosomes in lobule 3 of the cerebellar vermis in wildtype (WT) and SCA6 mice, calbindin labels Purkinje cells. Scale bar, 100 μm. (**c**) Closeup images of the early endosome marker EEA1 in lobule 3 of cerebellum, calbindin labels

*Figure 2 continued on next page*

*Figure 2 continued*

Purkinje cells (outlined). Scale bar, 10 µm. (**d**) The area covered by EEA1 staining in Purkinje cells is increased in SCA6 mice compared to WT (p < 0.001; *N* = 6 WT mice, 173 cells; 5 SCA6 mice, 171 cells). (**e**) Representative images of the early endosome marker EEA1 and BDNF within Purkinje cells (outlined) in the anterior vermis. Scale bar, 10 µm. (**f**) Relative staining level of BDNF within early endosome compartment is higher in the Purkinje cells of SCA6 mice compared to WT (p < 0.001; *N* = 6 WT mice, 173 cells; 5 SCA6 mice, 171 cells). (**g**) Representative images of the early endosome marker EEA1 and TrkB within Purkinje cells (outlined) in the anterior vermis. Arrowheads denote locations of significant TrkB accumulations. Scale bar, 10 µm. (**h**) Relative staining level of TrkB within early endosome compartment is higher in the Purkinje cells of SCA6 mice compared to WT (p = 0.013; *N* = 4 WT mice, 79 cells; 4 SCA6 mice, 58 cells). Mann–Whitney *U*-test used for all statistical comparisons, *p < 0.05, ***p < 0.001.

two markers. The lysosomes labeled by these two strategies could be divided into three groups. (1) Compartments stained by both Lamp1 and LysoTracker, confirming that they are indeed lysosomes. The number of these lysosomes was indistinguishable between WT and SCA6 mice (*Figure 5f*), indicating that Lamp1-positive lysosomes are present in normal quantities and are appropriately acidified in SCA6 Purkinje cells. In agreement, the area covered by each stain per lysosome, a proxy for the size of the lysosomes, was unchanged in SCA6 (*Figure 5—figure supplement 1*). (2) Lysosomes labeled by Lamp1 but not LysoTracker (*Figure 5g*) represent small lysosomes unable to retain the LysoTracker dye after fixation (*Chazotte, 2011*; *Song et al., 2008*). This number was not significantly different between WT and SCA6 mice and so Lamp1 appears to be present at normal levels and is appropriately localized to lysosomes in SCA6. (3) Organelles labeled by LysoTracker but not Lamp1 could include maturing late endosomes or other endosomal compartments that are inappropriately acidified. Late endosomes have an acidic pH and Rab7-positive late endosomes can be LysoTracker positive (*Majzoub et al., 2016*). Late endosomes fuse with lysosomes to complete cargo degradation, but first they mature into an intermediate compartment called an endolysosome (*Huotari and Helenius, 2011*). Therefore, these LysoTracker-positive Lamp1-negative organelles could be late endosomes or intermediate endolysosomes that have not yet fused with the Lamp1-positive lysosomes. The number of these compartments was low in Purkinje cells in both genotypes but was significantly lower in SCA6 mice (*Figure 5h*). This agrees with our finding of a reduction in the size of the late endosome compartment in SCA6 mice (*Figure 4c*). A reduction in the late endosome compartment suggests that there are fewer late endosomes available to mature and fuse with lysosomes. Therefore, the Lamp1-positive lysosomes are morphologically normal in the Purkinje cells of SCA6 mice while the small alteration in Lysotracker staining likely represents a deficit in the maturation of late endosomes before fusion with lysosomes.

Taken together, these results suggest that although lysosomes are present in normal numbers, the trafficking of endocytosed cargo to the lysosome for degradation is impaired in the SCA6 cerebellum. Our transcriptomic data furthermore show that genes encoding lysosomal proteins are abnormally expressed in SCA6 (*Figure 1b*), suggesting that lysosomal dysfunction is present in SCA6. Lysosomes are enzymatically active sites of degradation and contain multiple proteases that rapidly break down proteins. However, lysosomal proteolysis depends not just on enzymatic activity, but on the correct transport of proteins to the lysosomes through endosomal compartments. We wondered whether the deficits we identified in early and late endosomes (*Figures 2 and 4*), or other deficits in lysosome activity, might impair the ability of lysosomes to degrade endocytosed cargo in SCA6. We assayed this by incubating acute cerebellar slices with BODIPY FL Pepstatin A, an exogenous peptide that can be endocytosed and transported to the lysosome, where it binds to the active form of the lysosomal enzyme Cathepsin D (*Chen et al., 2000*; *Figure 6a*). We could visualize Pepstatin A signal within Lamp1-positive puncta, demonstrating Cathepsin D enzymatic activity within lysosomes (*Figure 6b*). The Pepstatin A signal within Purkinje cells was significantly reduced in SCA6 mice (*Figure 6c, d*), revealing a lower level of either the endocytosed construct, or of active Cathepsin D. The identification of alterations in the early and late endosomes that transport cargo to the lysosomes (*Figures 2 and 4*), suggests that SCA6 Purkinje cells have endosomal deficits that reduce transport of the endocytosed Pepstatin A construct to the lysosome where it can bind active Cathepsin D. This finding is therefore consistent with an overall impairment of endo-lysosomal trafficking. The reduction in Pepstatin A signal would also be consistent with a reduction in Cathepsin D enzymatic activity that could further contribute to aberrant handling of endocytosed cargo in SCA6. RNA-sequencing identified an upregulation of lysosomal enzyme transcripts in the SCA6 cerebellum, including the *CTSD* gene encoding Cathepsin D (*Figure 1b*). However, this would not necessarily lead to an increase in

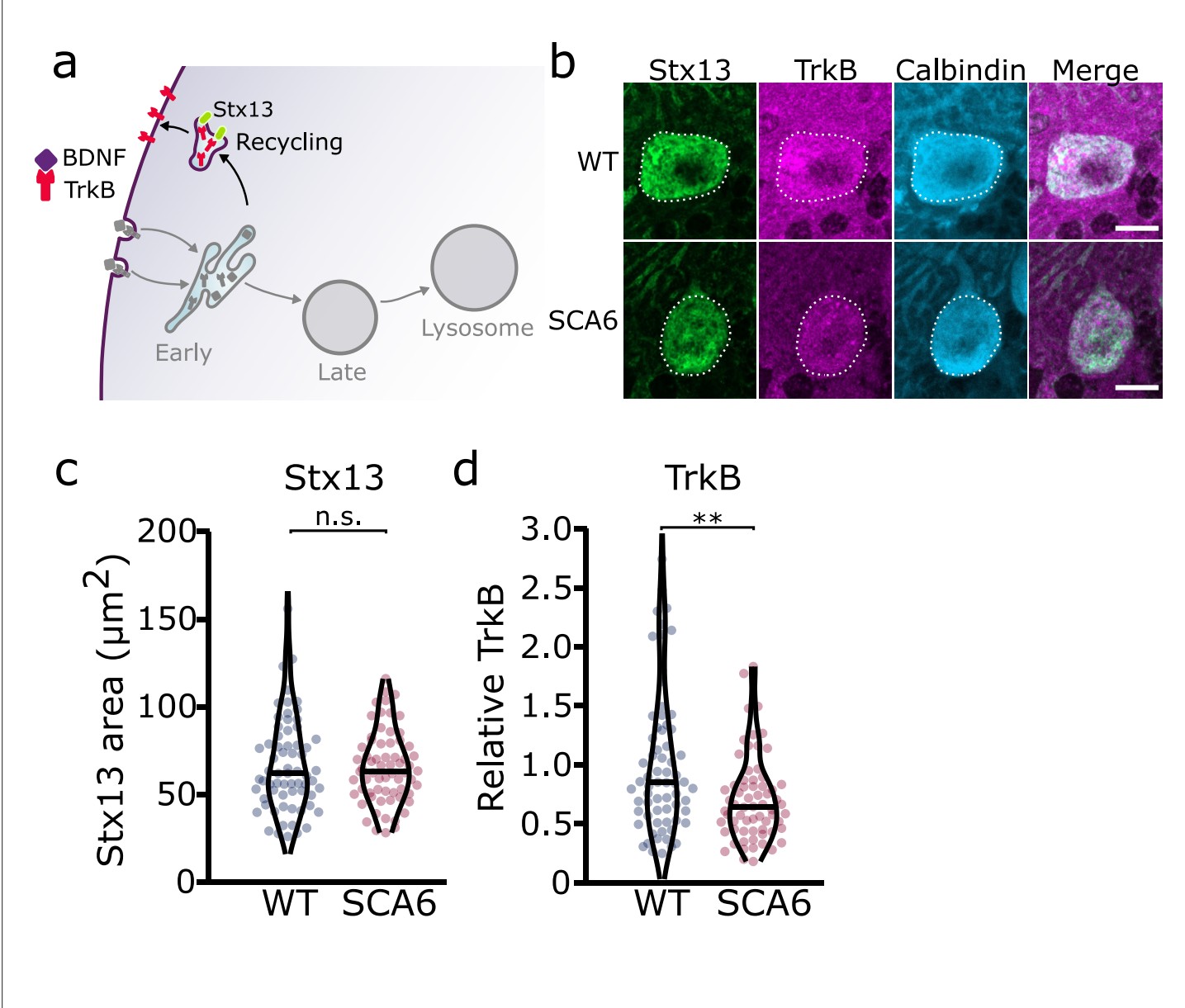

**Figure 3.** Recycling of tropomyosin kinase B (TrkB) in recycling endosomes is impaired in spinocerebellar ataxia type 6 (SCA6) Purkinje cells. (**a**) Schematic showing the return of TrkB to the cell membrane via recycling endosomes labeled with Stx13. (**b**) Representative images of the recycling endosome marker Stx13 and TrkB within Purkinje cells (outlined) of the anterior vermis. Scale bar, 10 μm. (**c**) The area covered by Stx13 staining in Purkinje cells is unchanged between wildtype (WT) and SCA6 mice (p = 0.96; *N* = 4 WT mice, 67 cells; 4 SCA6 mice, 66 cells). (**d**) The relative level of TrkB within recycling endosomes was significantly decreased in SCA6 Purkinje cells (p = 0.0034; *N* = 4 WT mice, 67 cells; 4 SCA6 mice, 66 cells). Mann–Whitney *U*-test used for all statistical comparisons, **p < 0.01, n.s. p > 0.05.

The online version of this article includes the following figure supplement(s) for figure 3:

**Figure supplement 1.** A second Stx13 antibody confirms that recycling endosome area is unchanged in spinocerebellar ataxia type 6 (SCA6) Purkinje cells.

enzyme activity as the RNA must first be translated, and then undergo post-translational proteolytic processing in order to produce the active form of Cathepsin D that binds Pepstatin A (*Zaidi et al., 2008*). This suggests that in SCA6, there is an upregulation in *CTSD* and other lysosomal genes that does not lead to an increase in functional enzymes. This may be a compensatory mechanism for trafficking deficits of cargoes to the lysosomal compartment, or an attempt to compensate for deficits in the production of the active form of Cathepsin D. In summary, our results illustrate that Purkinje cells

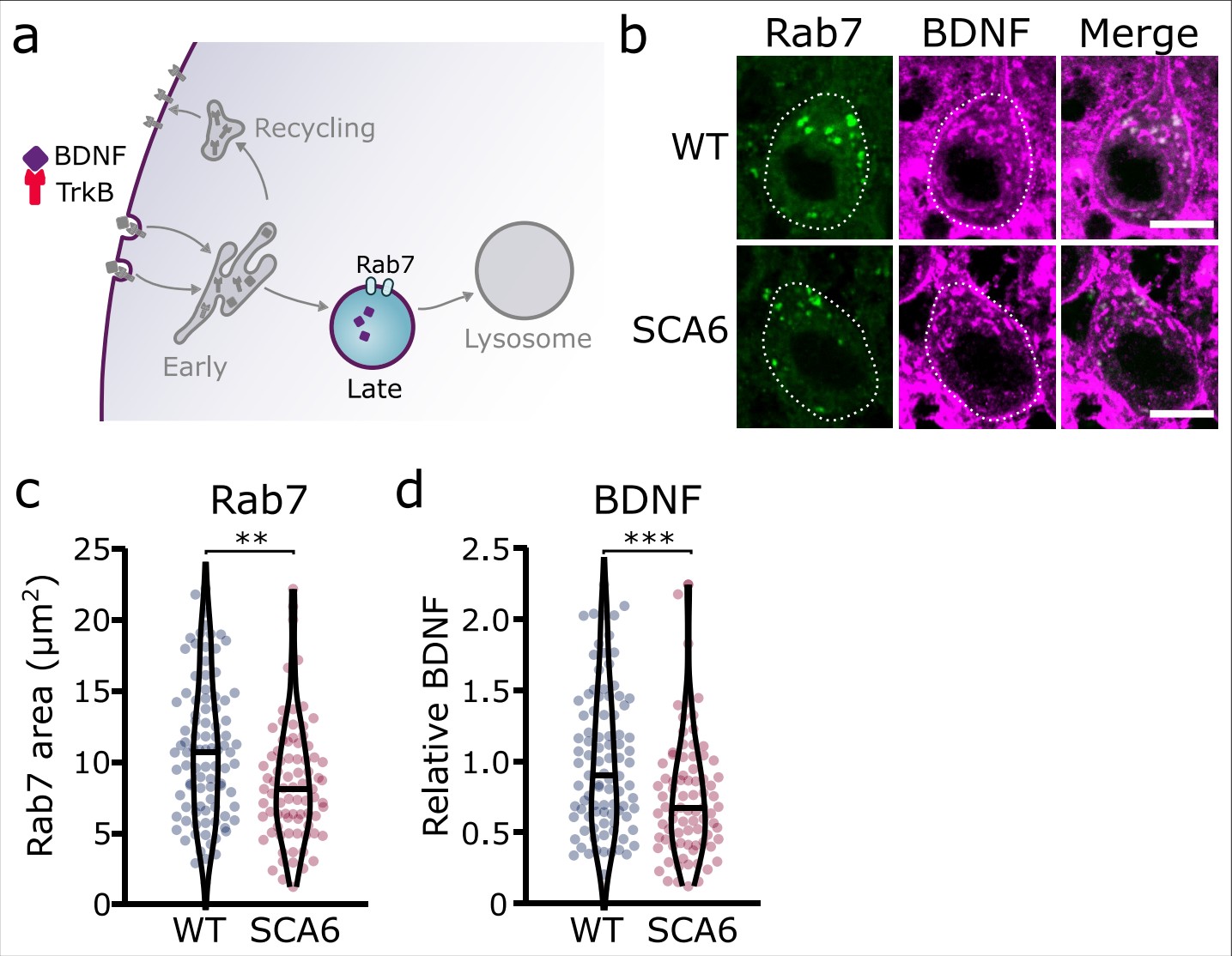

**Figure 4.** Late endosomes are reduced in spinocerebellar ataxia type 6 (SCA6) Purkinje cells and carry less brain-derived neurotrophic factor (BDNF). (**a**) Schematic showing the transport of BDNF in late endosomes expressing Rab7. (**b**) Representative images of the late endosome marker Rab7 and BDNF within Purkinje cells (outlined) of the anterior vermis. Scale bar, 10 μm. (**c**) The area covered by Rab7 staining in Purkinje cells is decreased in SCA6 mice compared to wildtype (WT) (p = 0.0019; N = 4 WT mice, 93 cells; 4 SCA6 mice, 81 cells). (**d**) The relative level of BDNF within late endosomes was significantly decreased in SCA6 Purkinje cells (p < 0.001; N = 4 WT mice, 93 cells; 4 SCA6 mice, 81 cells). Mann–Whitney *U*-test used for all statistical comparisons, **p < 0.01, ***p < 0.001.

in SCA6 mice have an impaired ability to degrade other endocytosed cargo beyond BDNF and TrkB. This likely arises from trafficking deficits that result in a disruption in the transport of cargo to the lysosomes, along with lysosomal dysfunction.

## 7,8-DHF rescues deficits in early endosomes

We have previously shown that the putative TrkB agonist 7,8-DHF increases TrkB expression and activates Akt signaling downstream of TrkB, leading to a rescue of deficits in motor coordination and Purkinje cell firing in SCA6 mice (*Figure 7a*; *Cook et al., 2022*). One interpretation of this rescue is that the TrkB agonist compensates for reduced levels of BDNF in the SCA6 cerebellum, thereby activating downstream signaling cascades to restore cerebellar function. However, we have also previously shown that TrkB activation with 7,8-DHF led to increased levels of TrkB in the SCA6 cerebellum (*Cook et al., 2022*), suggesting that self-amplification of TrkB signaling may play a role in the rescue. 7,8-DHF may therefore not be simply replacing lost BDNF in SCA6. Indeed, there is some evidence

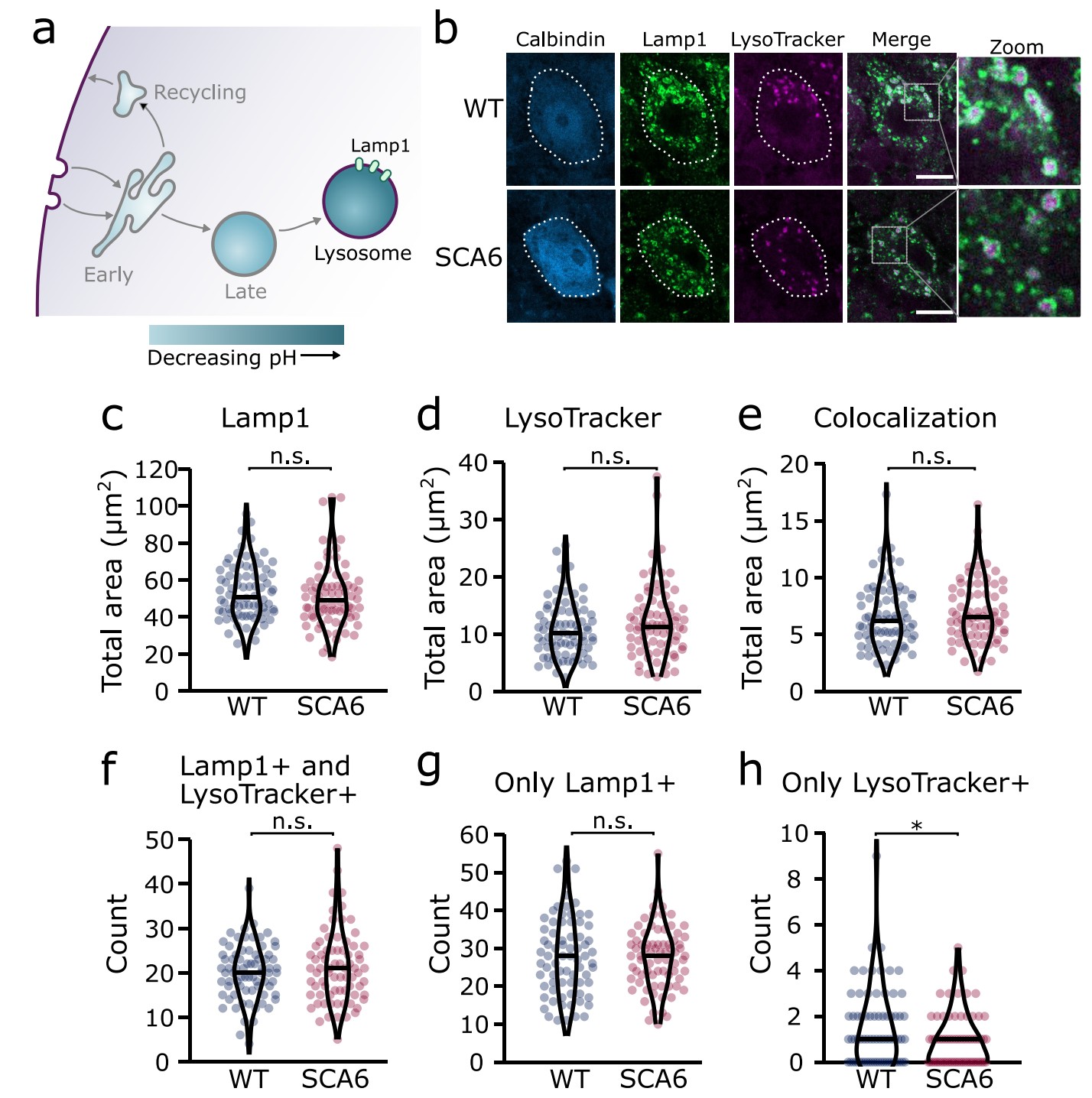

**Figure 5.** Lysosomes in spinocerebellar ataxia type 6 (SCA6) Purkinje cells are morphologically normal but there may be a deficit in late endosome maturation. (**a**) Schematic showing the endo-lysosomal system as a series of compartments of increasingly acidic pH with lysosomes, expressing Lamp1, being most acidic. (**b**) Representative images of Purkinje cells in the anterior vermis stained for the lysosome membrane marker Lamp1 and LysoTracker as a marker of acidic compartments. Calbindin labels Purkinje cells (outlined). Scale bar, 10 μm. The area covered by (**c**) Lamp1 staining and (**d**) LysoTracker staining in Purkinje cells is unchanged between wildtype (WT) and SCA6 mice (p = 0.26 for Lamp1; p = 0.20 for LysoTracker; $N$ = 3 WT mice, 78 cells; 3 SCA6 mice, 76 cells). (**e**) The area of colocalization between Lamp1 and Lysotracker is unchanged between WT and SCA6 mice (p = 0.34; $N$ = 3 WT mice, 78 cells; 3 SCA6 mice, 76 cells). (**f**) The number of lysosomes that were both Lamp1- and LysoTracker positive was not significantly different between genotypes (p = 0.72; $N$ = 3 WT mice, 78 cells; 3 SCA6 mice, 76 cells). (**g**) The number of lysosomes that were Lamp1 positive, but LysoTracker negative was unchanged between genotypes (p = 0.83; $N$ = 3 WT mice, 78 cells; 3 SCA6 mice, 76 cells). (**h**) The number of putative late

*Figure 5 continued on next page*

*Figure 5 continued*

endosomes undergoing maturation that were Lamp1 negative, but LysoTracker positive was significantly decreased in the Purkinje cells of SCA6 mice (p = 0.034; *N* = 3 WT mice, 78 cells; 3 SCA6 mice, 76 cells). Mann–Whitney *U*-test used for all statistical comparisons except number of Lamp1+ puncta (**f**) which was normally distributed and so a Student's *t*-test was used, *p < 0.05, n.s. p > 0.05.

The online version of this article includes the following figure supplement(s) for figure 5:

**Figure supplement 1.** Lysosome size is unchanged in spinocerebellar ataxia type 6 (SCA6) Purkinje cells.

that 7,8-DHF may act via antioxidant or anti-inflammatory effects independent of TrkB activation (*Chen et al., 2011*; *Han et al., 2014*; *Park et al., 2012*). Given that TrkB signaling is known to be involved in the regulation and trafficking of endosomes (*Scott-Solomon and Kuruvilla, 2018*), we wondered whether chronic oral administration of 7,8-DHF to SCA6 mice would rescue the deficits in early endosome size, either due to TrkB activation or another mechanism of action. We used the same treatment regimen of 7,8-DHF that had previously been shown to rescue deficits in behavior and Purkinje cell firing, which comprised of 1 month of oral administration of the drug to peri-onset (6- to 7-month-old) SCA6 mice (*Figure 7a*). Half of the mice received control solution with no drug added. After 1 month of 7,8-DHF administration, cerebellar vermis sections were stained for BDNF and EEA1 as an early endosome marker, as previously described (*Figure 2*). 7,8-DHF administration rescued the size of the early endosomes, with the total EEA1 area within Purkinje cells reducing to the WT level previously described (*Figure 7b, c*). The level of BDNF accumulation within the early endosomes was also restored (*Figure 7d*), suggesting that in addition to self-amplifying upregulation of TrkB (*Cook et al., 2022*), treatment with 7,8-DHF can normalize intracellular localization and trafficking of BDNF, consistent with the role of TrkB in regulating endosome trafficking and function (*Scott-Solomon and Kuruvilla, 2018*).

## Discussion

We have identified multiple novel deficits in the endo-lysosomal pathway that could contribute to BDNF–TrkB signaling alterations in SCA6 (*Figure 8*). Early endosomes were enlarged in SCA6 and accumulate BDNF and TrkB. Recycling endosomes were morphologically normal but carried less TrkB, and late endosomes were reduced in size and carried less BDNF. This suggests that cargo is getting stuck in the enlarged early endosomes in SCA6. The involvement of lysosomal pathology in SCA6 is less clear. Lysosomes appear morphologically normal with normal pH, but RNA-sequencing revealed upregulation of lysosomal genes in SCA6. SCA6 Purkinje cells also showed a reduction in exogenous cargo binding to the lysosomal enzyme Cathepsin D, indicating either a deficit in lysosomal enzyme activity, defects in endocytosis and cargo trafficking to lysosomes, or both. Taken together, these deficits could lead to cellular dysfunction in SCA6. This work further highlights how BDNF and TrkB deficits manifest in SCA6, and the potential of targeting this pathway therapeutically.

Excitingly, both the enlargement of early endosomes and the accumulation of BDNF in early endosomes could be rescued by the putative TrkB agonist, 7,8-DHF. There is evidence that TrkB can regulate its own expression and signaling (*Cheng et al., 2011*; *Haapasalo et al., 2002*) and we had previously shown that 7,8-DHF administration increased levels of TrkB in the cerebellum of SCA6 mice (*Cook et al., 2022*). This new finding demonstrates that not only does 7,8-DHF regulate the expression of TrkB in the SCA6 cerebellum, but also the localization of the ligand for TrkB within endosomal compartments, providing insight into the mechanisms by which BDNF and TrkB are regulated in the cerebellum. Furthermore, the finding that 7,8-DHF can rescue endosomal deficits in SCA6 supports the promise of 7,8-DHF and related molecules as potential therapeutics for SCA6 and raises the possibility of their use for treatment of other disorders displaying early endosome deficits.

Early endosome deficits are not unique to SCA6, meaning that this finding may have a wider application. Enlargement of early endosomes is a hallmark of dysfunction in the endosomal pathway (*Kaur et al., 2018*) and has been observed in multiple diseases including Alzheimer's disease (*Cataldo et al., 2004*; *Cataldo et al., 2000*; *Cataldo et al., 1997*; *Tate and Mathews, 2006*), Down syndrome (*Cataldo et al., 2000*), models of macular degeneration (*Kaur et al., 2018*), Charcot–Marie–Tooth (*Bucci et al., 2012*), and Parkinson's disease (*Vidyadhara et al., 2019*). Early endosome abnormalities can arise from deficits in maturation or fusion of early endosomes, or alterations in the microtubule networks that govern endosome fusion (*Skjeldal et al., 2012*). Purkinje cells may be particularly

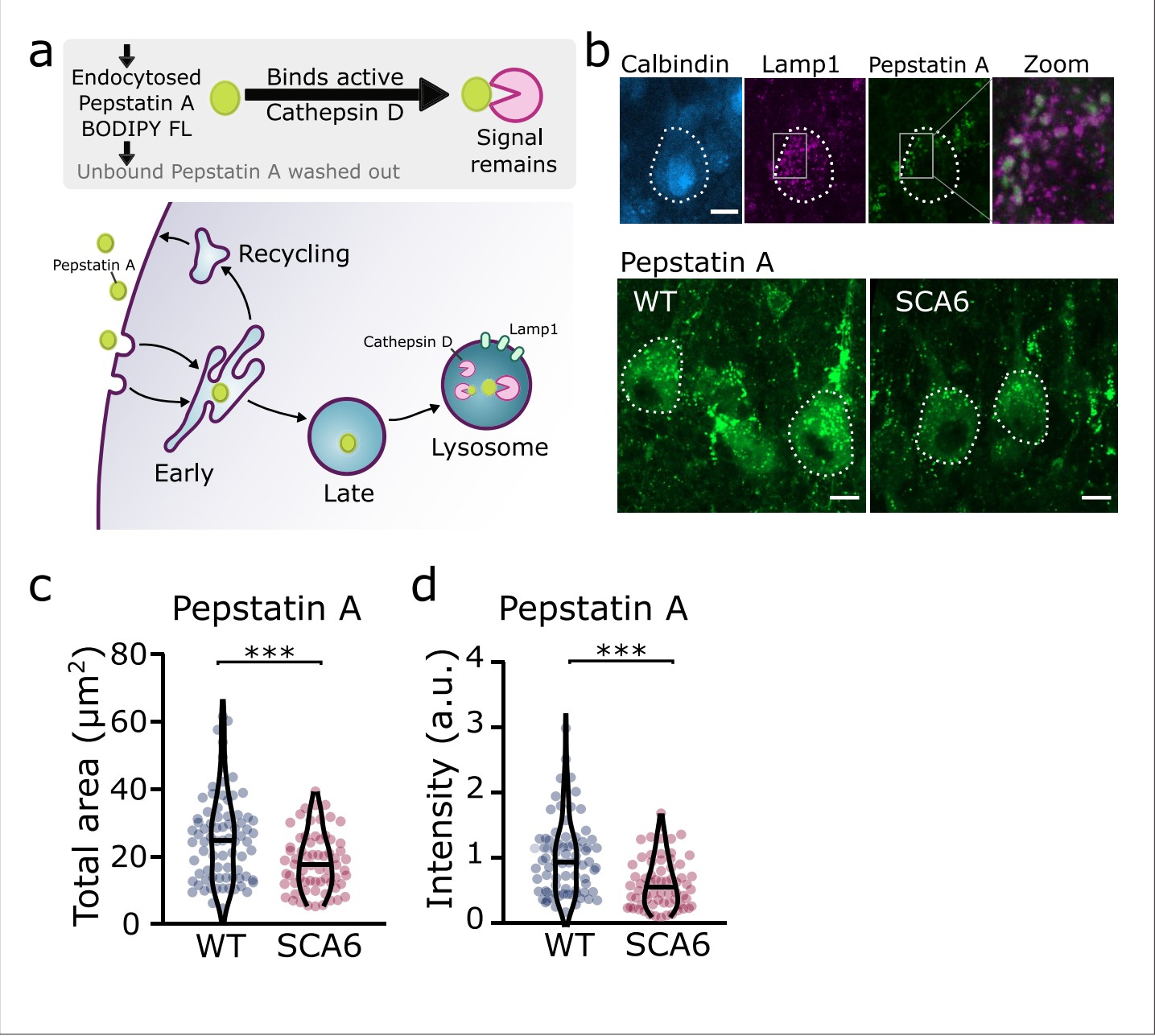

**Figure 6.** Trafficking of an exogenous peptide to the lysosome shows impairment in lysosomal action on endocytosed cargo in spinocerebellar ataxia type 6 (SCA6). (**a**) Schematic showing the use of Pepstatin A BODIPY FL construct to visualize endosome trafficking and Cathepsin D activity. (**b**) Representative images of Pepstatin A staining in the anterior vermis. Top row shows colocalization with lysosome marker Lamp1, Calbindin labels Purkinje cells (outlined). Scale bars, 10 µm (**c**) The area covered by Pepstatin A staining in Purkinje cells was significantly decreased in SCA6 mice (p = 0.00076; N = 3 wildtype (WT) mice, 78 cells; 3 SCA6 mice, 66 cells). (**d**) The intensity of Pepstatin A staining within Purkinje cells was significantly decreased in SCA6 mice (p < 0.0001; N = 3 WT mice, 78 cells; 3 SCA6 mice, 66 cells). ann–Whitney U-test used for all statistical comparisons, ***p < 0.001.

susceptible to early endosome enlargement, as in postmortem tissue from Niemann Pick type C patients, Purkinje cells had enlarged early endosomes which accumulated amyloid precursor protein, whereas in the hippocampus, pathology mostly concerned late endosomes (*Jin et al., 2004*). However, endosomal pathology in SCA6 and other diseases is not limited to early endosomes, and other endosomal compartments can be targeted therapeutically (*Bonam et al., 2019*; *Tate and Mathews, 2006*).

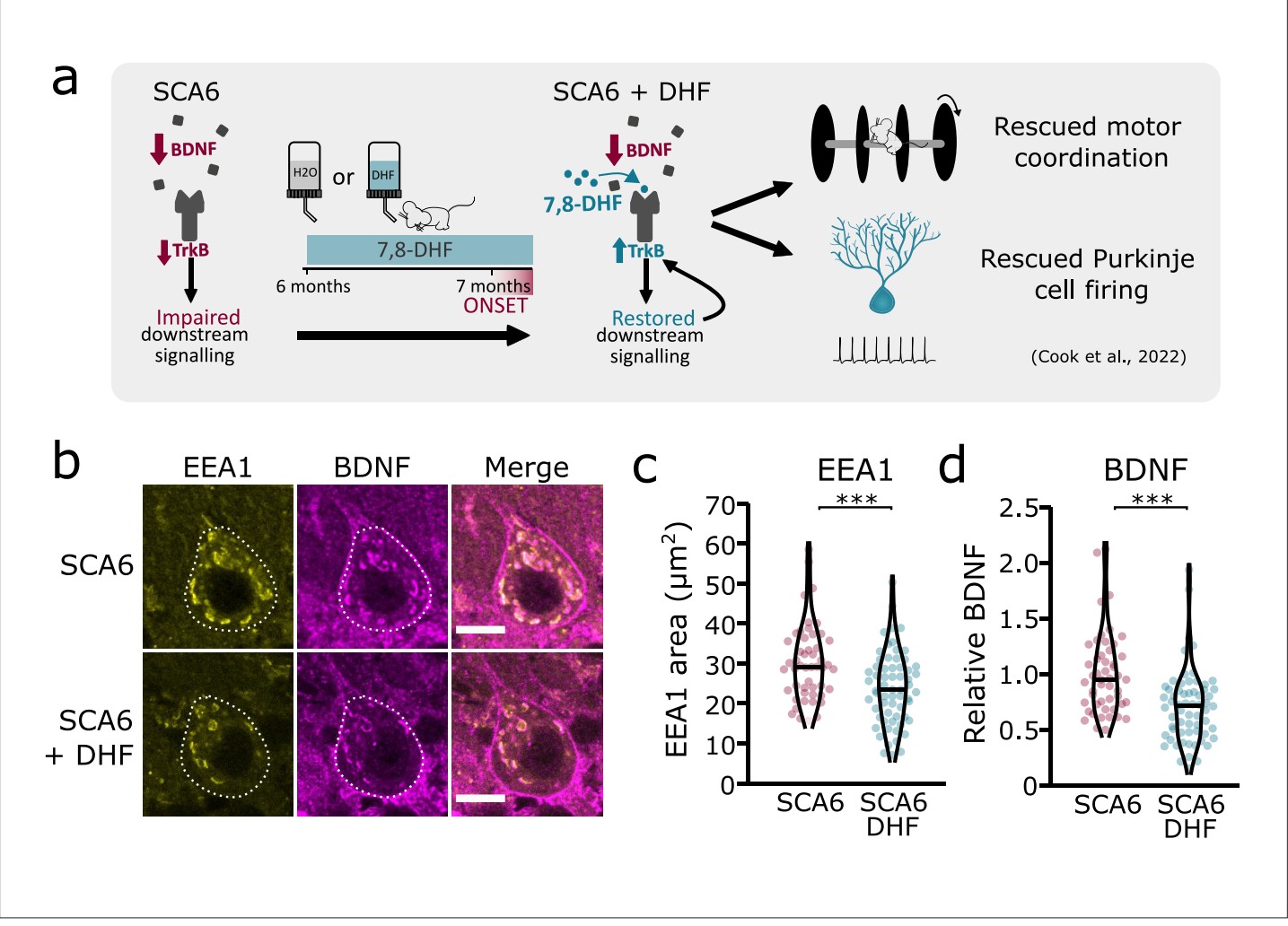

**Figure 7.** Tropomyosin kinase B (TrkB) activation with 7,8-dihydroxyflavone (7,8-DHF) rescues early endosomes deficits in spinocerebellar ataxia type 6 (SCA6) Purkinje cells. (**a**) Schematic showing 7,8-DHF administration to SCA6 mice and the previously described therapeutic benefits of 7,8-DHF in SCA6 mice. (**b**) Representative images of the early endosome marker EEA1 and brain-derived neurotrophic factor (BDNF) within Purkinje cells (outlined) of the anterior vermis. Scale bar, 10 μm. (**c**) The area covered by EEA1 staining in Purkinje cells is decreased in SCA6 mice given DHF compared to those that received control water (p < 0.001; Student's t-test; N = 3 SCA6 mice, 50 cells; 3 SCA6 DHF mice, 63 cells). (**d**) Relative staining level of BDNF within early endosome compartment is decreased in the Purkinje cells of SCA6 mice that received DHF compared to controls (p < 0.001; Mann–Whitney U-test; N = 3 SCA6 mice, 50 cells; 3 SCA6 DHF mice, 63 cells). ***p < 0.001.

In addition to identifying TrkB activation as a therapeutic strategy for endosome deficits, our work opens the door for the development of additional endosome-targeting treatment strategies for SCA6.

Our findings provide the first evidence for endosomal dysfunction contributing to signaling deficits in spinocerebellar ataxias. Other ataxias like SCA1 also have reduced BDNF protein levels (*Mellesmoen et al., 2018*), so it would be interesting to determine whether early endosome deficits could contribute to disrupted BDNF signaling in other ataxias. Polyglutamine diseases in general are thought to have disruption in protein degradation pathways (*Chai et al., 1999*; *Cortes and La Spada, 2015*), and SCA7 patients and mouse models display alterations in lysosome pathways (*Alves et al., 2014*). Therefore, similar abnormalities could be a common feature of similar diseases.

There remain, however, some unanswered questions from our study. What is the specific deficit that leads to early endosome enlargement and other deficits? Is it the same as in other neurological diseases? Future work to identify this will further aid in the development of endosome-targeting therapies. There is also the question of the effect on other cargo. Sorting and recycling happens in a cargo-specific manner (*Hu et al., 2015*) and so other cargo could be handled differently. Here, we

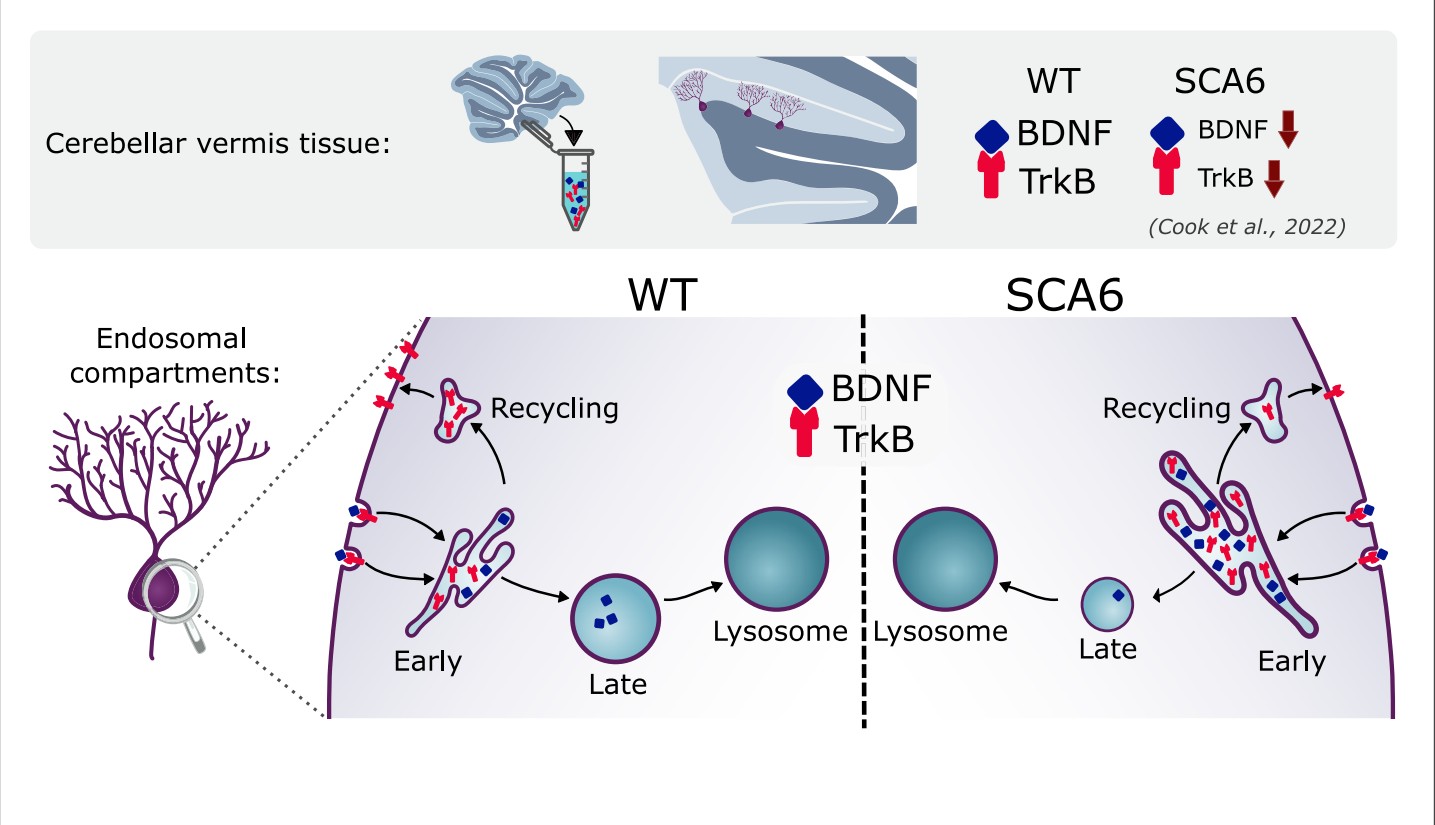

**Figure 8.** Multiple deficits in the endo-lysosomal system lead to abnormal localization of brain-derived neurotrophic factor (BDNF) and tropomyosin kinase B (TrkB) in the Purkinje cells of spinocerebellar ataxia type 6 (SCA6) mice.

investigated endosomal trafficking of BDNF and TrkB as we have previously shown this pathway to be both dysregulated in SCA6, and amenable to therapeutic targeting. However, our experiments with BODIPY FL Pepstatin A suggest that even exogenous cargo could be transported differently in SCA6. Future research should consider whether other cargo accumulate in early endosomes and how this affects cell physiology. Furthermore, in parallel to endocytosis and endosome trafficking, cells transport proteins for degradation in lysosomes via autophagy (*Klionsky, 2005*). There are therefore two pathways with a common destination of degradation in lysosomes, both of which are important for cell function, and abnormalities in one pathway can lead to disruption in the other (*Birgisdottir and Johansen, 2020*; *Mahapatra et al., 2019*). Future work should therefore investigate whether autophagy is disrupted in SCA6. Finally, while most of our study was conducted around disease onset, the transcriptomic alterations were measured at a later timepoint, which raises the question of whether transcriptional changes drive endo-lysosomal alterations or are in fact a result of endo-lysosomal changes that are caused by a different mechanism. A careful study of the timeline of these changes will help us understand the sequence of events as disease progresses in SCA6.

In summary, we have identified a dysregulated cellular trafficking pathway that contributes to cerebellar pathophysiology that is novel not just in SCA6, but in SCAs in general. We found that several components of the endo-lysosomal system show abnormalities in Purkinje cells in SCA6 mice. These deficits could contribute to BDNF–TrkB signaling deficits in SCA6, as BDNF and TrkB accumulate in early endosomes and there is a reduction in TrkB being recycled to the cell membrane for further signal transduction. The endo-lysosomal system is therefore a promising therapeutic target for SCA6.

# Materials and methods

## Animals

As previously described, the SCA6[84Q/84Q] mouse model is a knock-in mouse with a humanized 84 CAG repeat expansion mutation at the *CACNA1A* locus (*Jayabal et al., 2015*; *Watase et al., 2008*). We bred heterozygous mice to generate SCA6[84Q/84Q] and litter-matched WT control mice. Breeding mice were originally purchased from Jackson laboratories (Bar Harbor, Maine; strain: B6.129S7-Cacna1atm3Hzo/J; stock number: 008683) and primer sequences for genotyping were provided by Jackson laboratories. This breeding setup allowed us to use experimental cohorts that contained litter- and sex-matched mice randomly assigned to experimental groups. Both male and female mice were included in all experiments, and we did not exclude any experimental animals from analysis, in accordance with ARRIVE guidelines. The only experiment that included pre-selection of mice was for RNA-sequencing where mice were selected for tissue collection based on rotarod performance, to include animals that were representative of their genotype. For all other experiments mice were selected at random and none were excluded. Animal husbandry and procedures were approved by the McGill Animal Care Committee (protocol 6082) in accordance with the Canadian Council on Animal Care guidelines. All staining experiments were carried out on tissue from mice aged 7–8 months of age, except the Lamp1 and Lysotracker staining which was carried out at 12 months of age.

## RNA extraction and sequencing

RNA extraction was performed on tissues from five WT (two male, three female) and five SCA6 (three male, two female) mice at 12 months of age. Experimental mice were selected based on rotarod performance in order to identify mice that were phenotypically representative of their genotype. The experimental area and tools were cleaned with RNAseZap (Sigma #R2020) to remove RNAse contamination prior to starting experiments. Mice were anesthetized with isoflurane until the loss of toe pinch reflex, decapitated and the cerebellar vermis was dissected, separated into anterior and posterior sections, flash frozen on dry ice, and stored at −80°C until RNA extraction. RNA extraction proceeded as follows, with all procedures carried out on ice or at 4°C. Cerebellar vermis tissue sections were mechanically homogenized in TRIzol reagent (Invitrogen #15596026) by passing the lysate repeatedly through a series of needles, starting at 18-gauge, followed by 23-gauge. Chloroform (Sigma #288306) was added to the homogenate to separate total RNA in the aqueous phase, and RNA was recovered by precipitation with ethanol (70%). RNA extracts were purified using the RNeasy Mini Kit (QIAGEN #74104) with DNase I treatment as described in the manufacturer's protocol.

The RNA integrity of samples was quantified with capillary electrophoresis (Agilent BioAnalyzer 2100) in order to generate an RNA integrity number (RIN). All samples had an RIN over 7.5 and therefore were assessed to have high RNA quality. Library preparation and sequencing was carried out by Genome Quebec (Montreal, Quebec, Canada) using NEB stranded mRNA library prep, sequenced on the Illumina NovaSeq 6000 S4 generating 100 bp paired-end reads. Samples had a median of 53 million reads (Interquartile range [IQR], 42–64 million reads), and a median of 92% mapping rate (IQR, 92–92%).

## RNA-sequencing data analysis

The generated FASTQ files were transferred to the Digital Research Alliance of Canada cloud server for processing and analysis. First, reads were aligned to the mouse reference genome (GRCm38) using HISAT2 version 2.1.0 (*Kim et al., 2019*). Quantification of reads was carried out with HTSeq version 0.11. (*Anders et al., 2015*) in union mode. Reads generated from anterior and posterior samples were combined by addition at this step to generate an in silico whole cerebellar vermis read count. Differential expression analysis was performed with DESeq2 version 1.24.0 in RStudio (*Love et al., 2014*). DESeq2 adjusted p values for multiple testing with a target $a = 0.05$, and genes were considered to be differentially expressed genes at false discovery rate <0.05. Enriched gene pathways were identified using gProfiler (*Raudvere et al., 2019*) based on terms in GO databases.

## Immunohistochemistry

Tissue sections for IHC were prepared as described previously (*Cook et al., 2022*). Mice were deeply anesthetized with 400 mg/kg 2,2,2-tribromoethanol (Avertin) prepared in sterile 0.85% saline for a working concentration of 20 mg/ml. Intracardiac perfusion comprised of a flush with 40 ml of heparin

**Table 2.** List of antibodies.

| Antigen | Antibody (host) | Supplier | Catalog # | Dilution | Figure |
|---|---|---|---|---|---|
| EEA1 | Anti-EEA1 monoclonal (mouse) | Sigma | E7659 | 1:500 | 2, 7, 9 |
| BDNF | Anti-BDNF monoclonal (rabbit) | Abcam | ab108319 | 1:500 | 2, 4, 7, 9 |
| TrkB | Anti-TrkB polyclonal (rabbit) | EMD Millipore | ab9872 | 1:500 | 2, 3, 9 |
| Stx13 | Anti-Stx13 polyclonal (goat) | R&D Systems | AF6617 | 1:500 | 3 |
| Calbindin | Anti-calbindin polyclonal (guinea pig) | Synaptic Systems | 214 004 | 1:500 | 2, 3, 6, 9 |
| Rab7 | Anti-Rab7 monoclonal (mouse) | Cell Signaling Technology | 95746 | 1:400 | 4 |
| Calbindin | Anti-calbindin monoclonal (mouse) | Swant | 300 | 1:500 | 5 |
| Lamp1 | Anti-Lamp1 monoclonal (rat) | DSHB (University of Iowa)* | 1D4B | 1:400 | 5, 6 |
| Stx13 | Anti-Stx13 polyclonal (rabbit) | Synaptic systems | 110 133 | 1:150 | *Figure 3—figure supplement 1* |
| TrkB | Anti-TrkB polyclonal (rabbit) | Abcam | ab18987 | 1:500 | 9 |
| Anti-mouse secondary | Alexa 488 anti-mouse | Jackson Immunoresearch | AB_2338840 | 1:1000 | 2, 7, 9 |
| Anti-rabbit secondary | Alexa 594 anti-rabbit | Jackson Immunoresearch | AB_2340621 | 1:1000 | 2, 3, 7, 9 |
| Anti-mouse secondary | Alexa 594 anti-mouse | Jackson Immunoresearch | AB_2338871 | 1:1000 | 4 |
| Anti-rabbit secondary | Alexa 488 anti-rabbit | Jackson Immunoresearch | AB_2313584 | 1:1000 | 4 |
| SAnti-rat secondary | Alexa 488 anti-rat | Invitrogen | AB_2313584 | 1:1000 | 5 |
| Anti-goat secondary | Alexa 488 anti-goat | Jackson Immunoresearch | AB_2340428 | 1:1000 | 3 |
| Anti-mouse secondary | DyLight 405 anti-mouse | Invitrogen | 35501BID | 1:500 | 5 |
| Anti-guinea pig secondary | DyLight 405 anti-goat | Jackson Immunoresearch | AB_2340426 | 1:500 | 2, 3, 6, 9 |

*The LAMP-1 monoclonal antibody was obtained from the Developmental Studies Hybridoma Bank, created by the NICHD of the NIH and maintained at The University of Iowa, Department of Biology, Iowa City, IA 52242. It was deposited to the DSHB by August, J.T.

in phosphate-buffered saline (PBS; 0.1 M, pH 7) followed by 40 ml of 4% paraformaldehyde (PFA) in phosphate buffer (PB). Brains were rapidly removed and incubated in 4% PFA for a further 24 hr of fixation at 4°C on an orbital shaker at 70 RPM. Brains were dissected to separate the cerebellum and sliced using a vibrating microtome (5100mz Vibrotome, Campden Instruments, Loughborough, UK) into sections of 75 or 100 μm thickness. Slices were kept at 4°C in PBS with 0.05% sodium azide to inhibit microbial growth before staining.

Free-floating slices were rinsed with PBS and incubated in primary antibody (*Table 2*) and blocking solution with bovine serum albumin for 3 days, then rinsed in PBS and incubated for 90 min with secondary antibodies (*Table 2*). For experiments involving anti-mouse secondary antibodies, an additional incubation with Fab fragments (AffiniPure Fab Fragment Anti-Mouse IgG, Jackson Immunoresearch 715-007-003) was added prior to secondary antibody incubation. After a final PBS wash, stained slices were mounted using ProLong Gold Antifade mounting medium (Thermo Fisher Scientific, Waltham, USA) and stored at 4°C and protected from light prior to imaging.

As previously described, we found that the anti-BDNF antibody required epitope retrieval procedures to be carried out prior to staining in order to obtain adequate staining. We additionally found that epitope retrieval enhanced staining with anti-EEA1. Epitope retrieval took the form of a heat-induced epitope retrieval (HIER) step involving heating the slices in PBS to 95°C for 10 min immediately prior to staining. Slices were allowed to cool and IHC procedures were carried out as normal. This greatly enhanced the quality of BDNF and EEA1 staining and allowed us to resolve intracellular endolysosomal structures in Purkinje cells (*Figure 9a*). HIER did not affect the staining of Purkinje cells with anti-calbindin (*Figure 9a*) and so the effect was specific to certain antigens and antibodies and not to the tissue in general.

To further validate BDNF staining, we used liver tissue as a negative control. In rodent liver, the majority of tissue contains very low levels of BDNF (*Koppel et al., 2009*; *Vivacqua et al., 2014*), except from some BDNF-expressing cholangiocytes (*Vivacqua et al., 2014*). We therefore used liver tissue from a WT mouse undergoing surgery for another project as a negative control to validate the

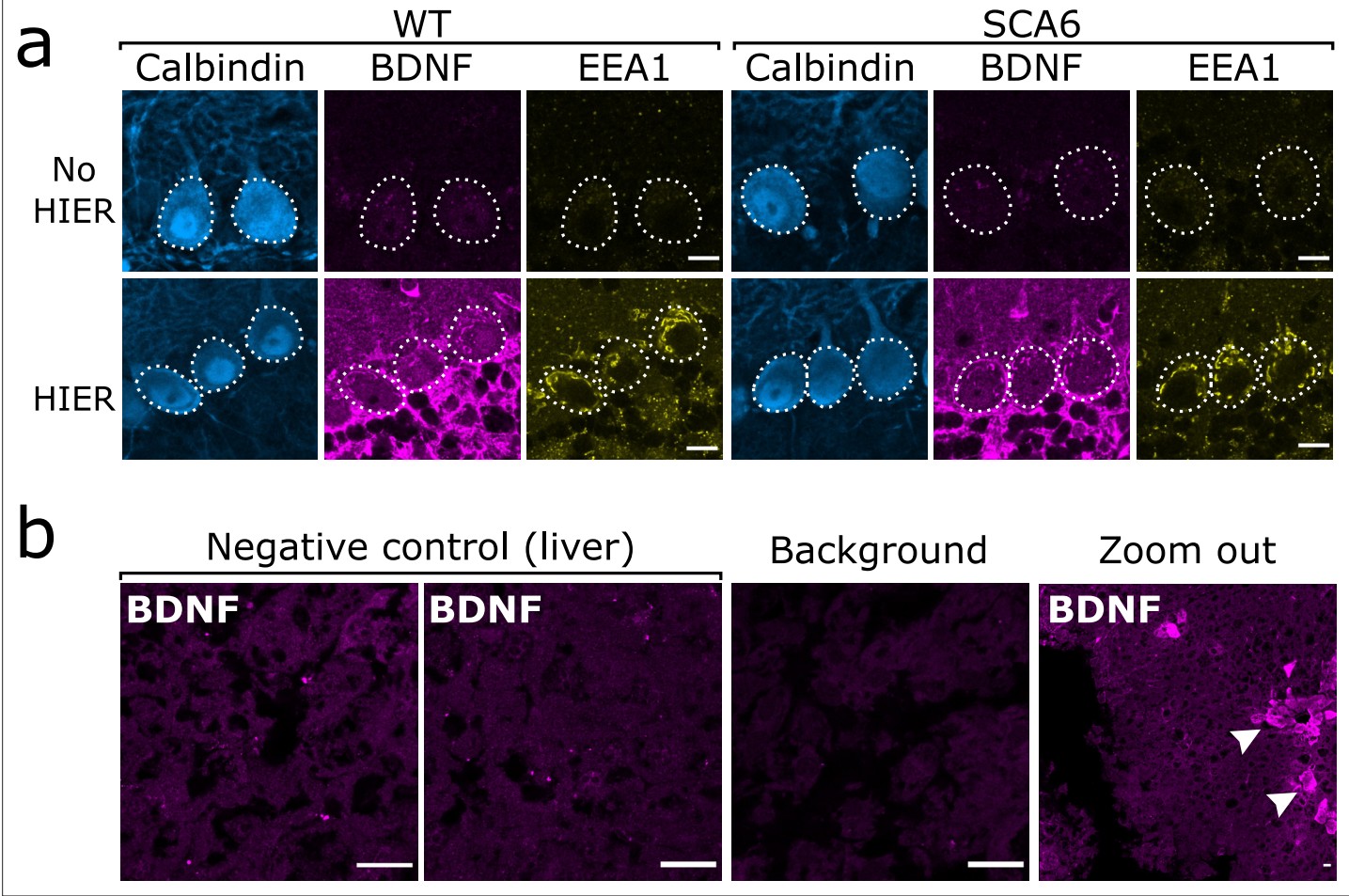

**Figure 9.** Brain-derived neurotrophic factor (BDNF) immunostaining in the cerebellum requires heat-induced epitope retrieval (HIER) and is not detectable in the majority of liver tissue. (**a**) HIER greatly enhances the signal from both BDNF and early endosome antigen 1 (EEA1) staining in cerebellar vermis tissue from wildtype (WT) and spinocerebellar ataxia type 6 (SCA6) mice. Calbindin staining is unaffected. Scale bars, 10 µm. (**b**) BDNF staining is undetectable in the majority of liver tissue from WT mouse, with only the BDNF-positive putative cholangiocytes (arrowheads) showing immunoreactivity with the BDNF antibody. Background slice was incubated without primary antibody but all other staining steps proceeded as normal. Scale bars, 20 µm.

The online version of this article includes the following figure supplement(s) for figure 9:

**Figure supplement 1.** A second tropomyosin kinase B (TrkB) antibody confirms endosomal localization of TrkB.

**Figure supplement 2.** Autofluorescence granules in Purkinje cells do not interfere with endosome identification.

BDNF antibody. The liver was recovered from the mouse during a terminal surgery and immediately placed into 4% PFA in PB for fixation. After 45 min of fixation the liver was sectioned into 100 µm slices using a vibrating microtome (5100mz Vibrotome, Campden Instruments, Loughborough, UK). These slices underwent HIER and BDNF staining as previously described. As expected, the majority of the liver tissue did not show BDNF immunoreactivity and we were able to detect putative cholangiocytes that were BDNF positive as previously described (*Figure 9b*).

To validate TrkB staining and confirm localization, a second TrkB antibody was used (*Table 2* for antibody information). We used cerebellar vermis tissue from a WT mouse to carry out TrkB staining using two different antibodies in the same batch and with all other reagents unchanged. We found that although the alternative antibody did not have as strong signal-to-noise ratio (*Figure 9—figure supplement 1*), the TrkB staining displayed a similar localization and both antibodies were able to label TrkB in early endosomes within Purkinje cells.

Purkinje cells sometimes contained autofluorescent granules that were visible under excitation from a 561- or 488-nm laser even without primary antibody incubation (*Figure 9—figure supplement*

2). In order to ascertain that these granules were not interfering with the identification of endosomes or lysosomes, staining batches included a negative control slice from each animal where applicable. These slices underwent all of the usual blocking steps except primary antibodies were not added, while secondary antibody incubation and mounting proceeded as usual. In the case of the Pepstatin A negative control, slices were incubated in artificial cerebrospinal fluid (ACSF) without Pepstatin A BODIPY FL, and all other steps proceeded as usual. We found that although there was some background fluorescence signal in Purkinje and other cells, the intensity of the signal was lower than the staining from the antibodies of interest and was unlikely to interfere significantly with image analysis. This can be seen in side-by-side comparisons of raw images from the same mice in *Figure 9*, *Figure 9—figure supplement 2*.

## Lysotracker

Lysotracker staining was carried out on acute cerebellar slices. Mice were anesthetized with isoflurane until the loss of toe pinch reflex and then were decapitated into ice cold ACSF (125 mM NaCl, 2.5 mM KCl, 2 mM $CaCl_2$, 1 mM $MgCl_2$, 1.25 mM $NaH_2PO_4$, 26 mM $NaHCO_3$, and 20 mM glucose, bubbled with 95% $O_2$ and 5% $CO_2$ (carbogen) to maintain pH at 7.3; osmolality ~316 mOsm). The brain was removed and dissected, and the cerebellar vermis was sectioned into 150 μm sagittal slices using a VT 1200S vibratome (Leica Microsystems, Wetzlar, Germany) in ice cold ACSF. Slices were transferred to a slice chamber filled with 37°C ACSF bubbled with carbogen for 40 min before staining. Slices were then incubated in LysoTracker Red DND-99 (Thermo Fisher L7528) that was dissolved in room temperature ACSF at a concentration of 1:3000. Incubation occurred at room temperature on an orbital shaker at 70 RPM. After 5 min of incubation the slices were transferred to 4% PFA in PB for post-fixation at 70 RPM for 1 hr. Slices were then washed in PBS and prepared for IHC as outlined above.

## Pepstatin A

Pepstatin A imaging was carried out on acute cerebellar slices prepared in the same way as for the Lysotracker imaging outlined above. After slices were kept in 37°C ACSF bubbled with carbogen for 40 min they were incubated in Pepstatin A fl BODIPY (Thermo Fisher P12271) that was prepared at a concentration of 1 μg/ml in room temperature ACSF. A stock solution of 1 μg/1 μl pepstatin A in Dimethylsulfoxide (DMSO) was prepared, to improve solubility, giving a final concentration of 0.001% DMSO in the incubation solution. Incubation lasted 30 min at room temperature on an orbital shaker at 70 RPM. Slices were then rinsed twice with PBS and transferred to 4% PFA for post-fixation at 70 RPM for 30 min. Slices either underwent further staining with IHC for Lamp1 and calbindin or were mounted onto slides immediately using ProLong Gold Antifade mounting medium (Thermo Fisher Scientific, Waltham, USA) and stored at 4°C and protected from light.

## Image acquisition

Imaging was performed using an LSM800 confocal microscope (Zeiss) equipped with Zen Blue software. Images were acquired using ×63 and ×40 objectives at 1024 × 1024-pixel resolution. Imaging was carried out by an experimenter who was blinded to the experimental condition of the slides. All imaging and analysis were carried out in lobule 3 of the anterior vermis, a region of the cerebellum that we have previously shown to display cellular deficits in SCA6 (*Cook et al., 2022*; *Jayabal et al., 2015*).

## Image analysis

Image analysis was performed in FIJI (ImageJ; US National Institutes of Health) (*Rueden et al., 2017*; *Schindelin et al., 2012*). Purkinje cell bodies were delineated by hand using calbindin or BDNF as a marker so that the Purkinje cell soma area could be isolated for further analysis. To calculate the area of the endosomes or lysosomes, an Otsu thresholding function of the endosomal marker was used. Despeckle and watershed functions were applied to create a region of interest consisting of the relevant endosomal compartment within the cell. These regions were used to calculate the size of different compartments, and in the analysis of lysosomes the area of colocalization between regions of interest was calculated. The regions of interest delineated with various markers were subsequently used as a mask to measure fluorescence intensity levels of the cargo of interest. The cargo level

was calculated using the integrated density measurement in ImageJ within the region of interest delineating the compartment of interest, and all values were normalized to the mean WT value. In *Figure 7*, values were normalized to the mean SCA6 control value. Analysis was carried out using custom ImageJ macros alongside manual verification.

For additional analysis of the lysosomes, we performed manual counts of Lamp1- and Lysotracker-positive organelles within individual Purkinje cells.

To improve the legibility of figure panels, linear adjustments of the brightness and contrast were applied to some representative image sets. Adjustments were kept to a minimum and identical adjustments were applied to images from all conditions, occurring after image analysis. Image analysis was performed on raw images by an experimenter blinded to the experimental condition of the slides.

## 7,8-DHF administration

7,8-DHF administration was carried out as previously described (*Cook et al., 2022*), as we had previously found this treatment regimen to be both well tolerated and effective at restoring TrkB–Akt signaling in SCA6 mice. Briefly, single-housed mice were provided with ad libitum access to drug solutions in place of their drinking water for 1 month. These solutions contained either 157.3 µM 7,8-DHF (TCI America) dissolved in 0.04% DMSO vehicle (Sigma, Oakville, ON, Canada) or 0.04% DMSO only as a control, dissolved in autoclaved water with 10% sucrose (Sigma, Oakville, ON, Canada) added to make the solutions palatable and disguise any taste of the drug. Solutions were prepared fresh at least twice weekly and replenished daily. It has been shown by us and others that oral administration of 7,8-DHF is sufficient to induce signaling changes in the brain (*Cook et al., 2022*; *Gao et al., 2016*; *Johnson et al., 2012*; *Liu et al., 2016*; *Parrini et al., 2017*; *Zhang et al., 2014*) and the drug has been shown to be orally bioavailable (*Liu et al., 2013*).

## Statistical analysis

Pairwise statistical comparisons were carried out to identify statistical differences between groups. In cases where the data were normally distributed, Student's *t*-tests were used, otherwise Mann–Whitney *U*-tests were used. The type of statistical test and the sample size is indicated in the figure legend for each plot. All violin plots show individual cells as data points, with the median displayed as a horizontal line.

## Acknowledgements

We would like to thank all members of the Watt lab for helpful discussions and feedback on experiments, particularly to Kim Gruver for feedback on an earlier draft of this manuscript. Thank you to Anne McKinney, Louis-Charles Masson, and Andy Gao for advice and guidance throughout the project. Thank you to Rose Bagot for technical advice regarding the RNA-sequencing experiment.

We are grateful to Lois Lau and Anika Holur for carrying out genotyping and to Tanya Koch and all CMARC staff for animal colony support. Imaging was carried out in the McGill University Advanced BioImaging Facility (ABIF), RRID:SCR_017697 and we thank ABIF staff members for technical support. This research was enabled in part by support provided by Calcul Québec (calculquebec.ca) and the Digital Research Alliance of Canada (alliancecan.ca). This work was funded by a CIHR project grant (PJT-153150) (AJW). Student support came from doctoral awards from the Fonds de recherche du Québec – Santé (AAC and TCSL), a MITACS training award (AAC), a McGill science undergraduate research award (MR), and a McGill Biology research award (ÉZD).

## Additional information

### Funding

| Funder | Grant reference number | Author |
| --- | --- | --- |
| Canadian Institutes of Health Research | PJT-153150 | Alanna Jean Watt |

| Funder | Grant reference number | Author |
|---|---|---|
| Canadian Institutes of Health Research | PJT-190151 | Alanna Jean Watt |
| Canadian Institutes of Health Research | PJT-190129 | Alanna Jean Watt |
| Fonds de Recherche du Québec - Santé | Doctoral Award | Anna A Cook Tsz Chui Sophia Leung |
| Mitacs | Training award | Anna A Cook |
| McGill University | Science undergraduate research award | Max Rice |
| McGill University | Biology research award | Élyse Zadigue-Dube |

The funders had no role in study design, data collection, and interpretation, or the decision to submit the work for publication.

### Author contributions

Anna A Cook, Conceptualization, Data curation, Formal analysis, Investigation, Methodology, Writing – original draft; Tsz Chui Sophia Leung, Conceptualization, Data curation, Software, Investigation, Visualization, Methodology, Writing – review and editing; Max Rice, Formal analysis, Investigation, Methodology, Writing – review and editing; Maya Nachman, Data curation, Investigation, Methodology, Writing – review and editing; Élyse Zadigue-Dube, Data curation, Formal analysis, Investigation, Methodology, Writing – review and editing; Alanna Jean Watt, Conceptualization, Supervision, Funding acquisition, Visualization, Writing – original draft, Project administration, Writing – review and editing

### Author ORCIDs

Anna A Cook http://orcid.org/0000-0001-7654-0212
Tsz Chui Sophia Leung https://orcid.org/0000-0003-2158-4300
Alanna Jean Watt https://orcid.org/0000-0002-6371-6220

### Ethics

Animal husbandry and procedures were approved by the McGill Animal Care Committee in accordance with the Canadian Council on Animal Care guidelines (protocol 6082).

Joint Public Review: https://doi.org/10.7554/eLife.90510.3.sa1
Author Response https://doi.org/10.7554/eLife.90510.3.sa2

## Additional files

### Supplementary files

• MDAR checklist

### Data availability

Source data files have been provided for Figure 1.

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
