## [Editor Report · eLife assessment]

This manuscript provides **valuable** insights to the underlying mechanism for Spinocerebellar ataxia 6 (SCA6) due to defective endolysosomal trafficking of BDNF and its receptor TrkB. The findings are **compelling** and significant in understanding the underlying pathology of SCA6. The authors have acknowledged the experimental weaknesses and recognize there may be multiple mechanisms to explain the findings.

---

## [Referee Report · Joint Public Review]

Cook, Watt, and colleagues previously reported that a mouse model of Spinocerebellar ataxia type 6 (SCA6) displayed defects in BDNF and TrkB levels at an early disease stage. Moreover, they have shown that one month of exercise elevated cerebellar BDNF expression and improved ataxia and cerebellar Purkinje cell firing rate deficits. In the current work, they attempt to define the mechanism underlying the pathophysiological changes occurring in SCA6. For this, they carried out RNA sequencing of cerebellar vermis tissue in 12-month-old SCA6 mice, a time when the disease is already at an advanced stage, and identified widespread dysregulation of many genes involved in the endo-lysosomal system. Focusing on BDNF/TrkB expression, localization, and signaling they found that, in 7-8 month-old SCA6 mice early endosomes are enlarged and accumulate BDNF and TrkB in Purkinje cells. Curiously, TrkB appears to be reduced in the recycling endosomes compartment, despite the fact that recycling endosomes are morphologically normal in SCA6. In addition, the authors describe a reduction in the Late endosomes in SCA6 Purkinje cells associated with reduced BDNF levels and a probable deficit in late endosome maturation.

Strengths:

The article is well written, and the findings are relevant for the neuropathology of different neurodegenerative diseases where dysfunction of early endosomes is observed. The authors have provided a detailed analysis of the endo-lysosomal system in SCA6 mice. They have shown that TrkB recycling to the cell membrane in recycling endosomes is reduced, and the late endosome transport of BDNF for degradation is impaired. The findings will be crucial in understanding underlying pathology. Lastly, the deficits in early endosomes are rescued by chronic administration of 7,8-DHF.

Weaknesses:

The specificity of BDNF and TrkB immunostaining requires additional controls, as it has been very difficult to detect immunostaining of BDNF.

The revised manuscript has included additional analysis using epitope retrieval and a negative liver control with the Abcam antibody against BDNF. An alternative antibody that may be considered for BDNF detection is from Icosagen AS. This antibody has been found to be effective for immunofluorescence and immunoblot purposes.

Two other issues were brought up in the initial review process--

1. One important concern about the conclusions is that the RNAseq experiment was conducted in 12-month-old SCA6 mice suggesting that the defects in the endo-lysosomal system may be caused by other pathophysiological events and, likewise, the impairment in BDNF signaling may also be indirect, as also noted by the authors. Indeed, Purkinje cells in SCA6 mice have an impaired ability to degrade other endocytosed cargo beyond BDNF and TrkB, most likely because of trafficking deficits that result in a disruption in the transport of cargo to the lysosomes and lysosomal dysfunction.

2. Moreover, the beneficial effects of 7,8-DHF treatment on motor coordination may be caused by 7,8-DHF properties other than the putative agonist role on TrkB. Indeed, many reservations have been raised about using 7,8-DHF as an agonist of TrkB activity. Several studies have now debunked (Todd et al. PlosONE 2014, PMID: 24503862; Boltaev et al. Sci Signal 2017, PMID: 28831019) or at the very least questioned (Lowe D, Science 2017: see Discussion: https://www.science.org/content/blog-post/those-compounds-aren-t-what-you-think-they-are Wang et al. Cell 2022 PMID: 34963057). Another interpretation is that 7,8-DHF possesses antioxidant activity and neuroprotection against cytotoxicity in HT-22 and PC12 cells, both of which do not express TrkB (Chen et al. Neurosci Lett 201, PMID: 21651962; Han et al. Neurochem Int. 2014, PMID: 24220540). Thus, while this flavonoid may have a beneficial effect on the pathophysiology of SCA6, it is most unlikely that mechanistically this occurs through a TrkB agonistic effect considering the potent anti-oxidant and anti-inflammatory roles of flavonoids in neurodegenerative diseases (Jones et al. Trends Pharmacol Sci 2012, PMID: 22980637).

---

## [Author Response]

The following is the authors’ response to the original reviews.

Cook, Watt, and colleagues previously reported that a mouse model of Spinocerebellar ataxia type 6 (SCA6) displayed defects in BDNF and TrkB levels at an early disease stage. Moreover, they have shown that one month of exercise elevated cerebellar BDNF expression and improved ataxia and cerebellar Purkinje cell ﬁring rate deﬁcits. In the current work, they attempt to deﬁne the mechanism underlying the pathophysiological changes occurring in SCA6. For this, they carried out RNA sequencing of cerebellar vermis tissue in 12-month-old SCA6 mice, a time when the disease is already at an advanced stage, and identiﬁed widespread dysregulation of many genes involved in the endo-lysosomal system. Focusing on BDNF/TrkB expression, localization, and signaling they found that, in 7-8 month-old SCA6 mice early endosomes are enlarged and accumulate BDNF and TrkB in Purkinje cells. Curiously, TrkB appears to be reduced in the recycling endosomes compartment, despite the fact that recycling endosomes are morphologically normal in SCA6. In addition, the authors describe a reduction in the Late endosomes in SCA6 Purkinje cells associated with reduced BDNF levels and a probable deﬁcit in late endosome maturation.

We would like to thank the reviewers for their careful reading of the paper, their feedback has helped us to add information and experiments to the paper that enhance the clarity of the ﬁndings.

Strengths:The article is well written, and the ﬁndings are relevant for the neuropathology of diﬀerent neurodegenerative diseases where dysfunction of early endosomes is observed. The authors have provided a detailed analysis of the endo-lysosomal system in SCA6 mice. They have shown that TrkB recycling to the cell membrane in recycling endosomes is reduced, and the late endosome transport of BDNF for degradation is impaired. The ﬁndings will be crucial in understanding underlying pathology. Lastly, the deﬁcits in early endosomes are rescued by chronic administration of 7,8-DHF.

We thank the reviewers for their positive feedback on this work.

Weaknesses:The speciﬁcity of BDNF and TrkB immunostaining requires additional controls, as it has been very diﬃcult to detect immunostaining of BDNF. In addition, in many of the ﬁgures, the background or outside of Purkinje cell boundaries also exhibits a positive signal.

We agree with the reviewers that the performance of the BDNF and TrkB antibodies is an important concern. We have ourselves had diﬃculties with the performance of many antibodies and the images in this paper are the result of many years of optimization. We have therefore added further detail about the antibody optimization to the methods section of this paper, and have carried out new staining experiments with additional controls. We have added 2 new ﬁgure panels in supplementary ﬁgures 3 and 4 to demonstrate these tests.

In the case of anti-BDNF antibodies, we have tested several antibodies and staining protocols and found that in our hands, the only antibody that reliably stained BDNF with a good signal to noise ratio was the one used in this paper (abcam ab108319). Even for this antibody, the staining was greatly enhanced by the use of a heat induced epitope retrieval (HIER) step, which allowed the visualization of BDNF within intracellular structures such as endosomes. When we quantiﬁed the intensity of this staining in our previous paper, the results were in agreement with those from a BDNF ELISA used to measure levels of BDNF in the cerebellar vermis of WT and SCA6 mice (Cook et al., 2022), which corroborates these results. As the staining was carried out in tissue sections and not dissociated cells, we also see positive signal from the BDNF staining outside of the Purkinje cells, since BDNF acts on cell-surface receptors and is thus released into the extracellular space around cells (Kuczewski et al., 2008) and is detectable in the extracellular matrix (Lam et al., 2019) and presynaptic terminals around neurons (Camuso et al., 2022; Choo et al., 2017). This is in contrast to studies that image BDNF mRNA with in-situ hybridization, which labels BDNF mRNA predominantly found in cells, and cannot tell us about sub-cellular or extracellular localization of BDNF protein. Together, these factors explain why we observe staining that is not cell- limited, but extends into the space around the cells of interest.

We have added an additional supplemental ﬁgure to demonstrate the importance of using HIER when staining slices with anti-BDNF (Supplementary ﬁgure 3). We tested HIER protocols that involved heating the slices to 95°C in a variety of buﬀers. The buﬀers tested were sodium citrate buﬀer (10 mM sodium citrate, 0.05% Tween 20, pH 6), Tris buﬀer (10mM TBS, 0.05% Tween 20, pH 10), EDTA buﬀer (1mM EDTA, 0.05% Tween 20, pH 8) and neutral PBS. The PBS produced the best result, enhancing the staining of both anti-BDNF and anti-EEA1 antibodies (Supplementary ﬁgure 3). Therefore all slices stained using those antibodies were heated to 95°C in PBS using a heat block or thermocycler for 10 minutes, then allowed to cool before staining proceeded.

The antibody we use (abcam ab108319) has been used in hundreds of other publications, including Javed et al., 2021 who ectopically expressed BDNF and noted colocalization between the antibody staining and the GFP tag of the BDNF construct, and Lejkowska et al., 2019 who overexpressed BDNF and saw a dramatic increase in antibody staining as well. The colocalization between ectopically expressed BDNF and the antibody in these studies demonstrates the speciﬁcity of the antibody.

However, to further validate antibody speciﬁcity we used liver tissue as a negative control. In liver tissue from rodents and humans, the majority of the liver contains negligible levels of BDNF (Koppel et al., 2009; Vivacqua et al., 2014), see also the Human Protein Atlas. The exception is some cholangiocytes: epithelial cells that express BDNF at high levels (Vivacqua et al., 2014). We obtained liver tissue from a WT mouse that was undergoing surgery for an unrelated project and ﬁxed and processed the tissue as we did for brain tissue (outlined in methods section). As we would expect, most of the cells in the liver showed BDNF immunoreactivity that was comparable to background levels (Supplementary ﬁgure 3). Interestingly, we were also able to detect sparse highly BDNF-positive cells in the liver, presumed cholangiocytes (Supp. Fig. 3). This pattern of liver BDNF expression is as predicted in the literature, and thus acts as a control for our antibody. We therefore believe that in our hands this antibody is able to stain BDNF with an appropriate degree of speciﬁcity.

We also carried out staining experiments using a second anti-TrkB antibody that we had previously used to detect TrkB via Western bloing. We carried out immunohistochemistry as previously described using tissue sections from a WT mouse. The staining with the two diﬀerent antibodies was carried out at the same time and all other reagents were kept constant. We found that both antibodies labelled TrkB in a similar pattern of localization, including in the early endosomes of the Purkinje cells (Supplementary ﬁgure 4). The second antibody however did have a lower signal to noise ratio and so we believe that the original anti-TrkB antibody used in this manuscript (EMD Millipore ab9872) is optimal for staining cerebellar tissue sections in our hands.

One important concern about the conclusions is that the RNAseq experiment was conducted in 12-month- old SCA6 mice suggesting that the defects in the endo-lysosomal system may be caused by other pathophysiological events and, likewise, the impairment in BDNF signaling may also be indirect, as also noted by the authors. Indeed, Purkinje cells in SCA6 mice have an impaired ability to degrade other endocytosed cargo beyond BDNF and TrkB, most likely because of traﬃcking deﬁcits that result in a disruption in the transport of cargo to the lysosomes and lysosomal dysfunction.

We agree with the reviewers that the defects in the endo-lysosomal system may be caused by other events occurring in the course of disease progression. As mentioned by the reviewers, we have noted this possibility in the text. Detailed investigation into the sequence of events and the root causes of signaling disruption in SCA6 merits future study and we aim to address this in future work. We have expanded this explanation in the text.

Moreover, the beneﬁcial eﬀects of 7,8-DHF treatment on motor coordination may be caused by 7,8-DHF properties other than the putative agonist role on TrkB. Indeed, many reservations have been raised about using 7,8-DHF as an agonist of TrkB activity. Several studies have now debunked (Todd et al. PlosONE 2014, PMID: 24503862; Boltaev et al. Sci Signal 2017, PMID: 28831019) or at the very least questioned (Lowe D, Science 2017: see Discussion: https://www.science.org/content/blog-post/those-compounds-aren-t-what-you-think-they-are Wang et al. Cell 2022 PMID: 34963057). Another interpretation is that 7,8-DHF possesses antioxidant activity and neuroprotection against cytotoxicity in HT-22 and PC12 cells, both of which do not express TrkB (Chen et al. Neurosci Lett 201, PMID: 21651962; Han et al. Neurochem Int. 2014, PMID: 24220540). Thus, while this ﬂavonoid may have a beneﬁcial eﬀect on the pathophysiology of SCA6, it is most unlikely that mechanistically this occurs through a TrkB agonistic eﬀect considering the potent anti-oxidant and anti-inﬂammatory roles of ﬂavonoids in neurodegenerative diseases (Jones et al. Trends Pharmacol Sci 2012, PMID: 22980637).

We thank the reviewers for raising this important point. We have noted in our previous paper (Cook et al., 2022) that 7,8-DHF may not be acting as a TrkB agonist in SCA6 mice, and are in agreement that other explanations are possible. We have now added information to the text of this paper to highlight this possibility. We did show in our previous paper that 7,8-DHF administration activates Akt signaling in the cerebellum of SCA6 mice, a signaling event that is known to take place downstream of TrkB activation. Additionally, 7,8-DHF treatment led to the increase of TrkB levels in the cerebellum of SCA6 mice (Cook et al., 2022), implicating TrkB in the mechanism of action, even if mechanistically, this is not via direct TrkB activation alone. However, even if the mechanism is currently incompletely explained, we believe that 7,8- DHF remains a valuable treatment strategy for SCA6. We have tried to rewrite the Discussion to highlight what we think is the most important takeaway: that 7,8-DHF can rescue endosomal and other deﬁcits in SCA6, even if we do not currently know the full mechanism of action. We have therefore amended the text to add more detail about other potential explanations for the mechanism of action of 7,8-DHF.

References

Camuso S, La Rosa P, Fiorenza MT, Canterini S. 2022. Pleiotropic eﬀects of BDNF on the cerebellum and hippocampus: Implications for neurodevelopmental disorders. Neurobiol Dis. doi:10.1016/j.nbd.2021.105606

Choo M, Miyazaki T, Yamazaki M, Kawamura M, Nakazawa T, Zhang J, Tanimura A, Uesaka N, Watanabe M, Sakimura K, Kano M. 2017. Retrograde BDNF to TrkB signaling promotes synapse elimination in the developing cerebellum. Nat Commun 8:195. doi:10.1038/s41467-017-00260-w

Cook AA, Jayabal S, Sheng J, Fields E, Leung TCS, Quilez S, McNicholas E, Lau L, Huang S, Watt AJ. 2022. Activation of TrkB-Akt signaling rescues deﬁcits in a mouse model of SCA6. Sci Adv 8:3260. doi:10.1126/sciadv.abh3260

Javed S, Lee YJ, Xu J, Huang WH. 2021. Temporal dissection of Rai1 function reveals brain-derived neurotrophic factor as a potential therapeutic target for Smith-Magenis syndrome. Hum Mol Genet 31:275–288. doi:10.1093/HMG/DDAB245

Koppel I, Aid-Pavlidis T, Jaanson K, Sepp M, Pruunsild P, Palm K, Timmusk T. 2009. Tissue-speciﬁc and neural activity-regulated expression of human BDNF gene in BAC transgenic mice. BMC Neurosci 10:68. doi:10.1186/1471-2202-10-68

Kuczewski N, Porcher C, Ferrand N, Fiorentino H, Pellegrino C, Kolarow R, Lessmann V, Medina I, Gaiarsa JL. 2008. Backpropagating action potentials trigger dendritic release of BDNF during spontaneous network activity. J Neurosci 28:7013–7023. doi:10.1523/JNEUROSCI.1673-08.2008

Lam D, Enright HA, Cadena J, Peters SKG, Sales AP, Osburn JJ, Soscia DA, Kulp KS, Wheeler EK, Fischer NO. 2019. Tissue-speciﬁc extracellular matrix accelerates the formation of neural networks and communities in a neuron-glia co-culture on a multi-electrode array. Sci Rep 9. doi:10.1038/s41598- 019-40128-1

Lejkowska R, Kawa MP, Pius-Sadowska E, Rogińska D, Łuczkowska K, Machaliński B, Machalińska A. 2019. Preclinical Evaluation of Long-Term Neuroprotective Eﬀects of BDNF-Engineered Mesenchymal Stromal Cells as Intravitreal Therapy for Chronic Retinal Degeneration in Rd6 Mutant Mice. Int J Mol Sci 2019, Vol 20, Page 777 20:777. doi:10.3390/IJMS20030777

Vivacqua G, Renzi A, Carpino G, Franchitto A, Gaudio E. 2014. Expression of brain derivated neurotrophic factor and of its receptors: TrKB and p75NT in normal and bile duct ligated rat liver. Ital J Anat Embryol 119:111–129. doi:10.13128/IJAE-15138